# Maternal vitamin B1 is a determinant for the fate of primordial follicle formation in offspring

Wen-Xiang Liu [1], Hai-Ning Liu[2], Zhan-Ping Weng[3], Qi Geng[1], Yue Zhang[1], Ya-Feng Li[1], Wei Shen[4], Yang Zhou [1] ✉ & Teng Zhang [1] ✉

The mediation of maternal-embryonic cross-talk via nutrition and metabolism impacts greatly on offspring health. However, the underlying key interfaces remain elusive. Here, we determined that maternal high-fat diet during pregnancy in mice impaired preservation of the ovarian primordial follicle pool in female offspring, which was concomitant with mitochondrial dysfunction of germ cells. Furthermore, this occurred through a reduction in maternal gut microbiota-related vitamin B1 while the defects were restored via vitamin B1 supplementation. Intriguingly, vitamin B1 promoted acetyl-CoA metabolism in offspring ovaries, contributing to histone acetylation and chromatin accessibility at the promoters of cell cycle-related genes, enhancement of mitochondrial function, and improvement of granulosa cell proliferation. In humans, vitamin B1 is downregulated in the serum of women with gestational diabetes mellitus. In this work, these findings uncover the role of the non-gamete transmission of maternal high-fat diet in influencing offspring oogenic fate. Vitamin B1 could be a promising therapeutic approach for protecting offspring health.

The worldwide prevalence of obesity and type 2 diabetes is increasing among couples of child-bearing age[1–3]. In addition, emerging evidence indicates that maternal obesity affects pregnancy outcome and offspring health[4,5]. Current literature aims to uncover the mechanisms that affect offspring phenotype in response to maternal obesity, and has found that intergenerational effects are mediated by maternal-embryonic cross-talk, which potentially leads to lifelong consequences[6,7].

Growth and development of the mammalian fetus and neonate is constrained by the nutrient supply from their mother; in turn, these nutrients are liberated and metabolized by maternal microorganisms living in the gut[8,9]. Thus, maternal microbial metabolites and xeno-biotics play pivotal roles in the modulation of maternal-fetal home-ostasis and offspring development[9–11]. It is known that the maternal gut microbiome regulates early embryonic brain development and

offspring's behavior in later-life[12]. A recent study found that maternal obesity induces cognitive and social behavioral deficits in the offspring of humans and mice, mediated by the mother-to-offspring transmission of the gut microbiome[13]. Additionally, the maternal gut micro-biota also influences the metabolic phenotype of offspring[14,15].

The mechanisms underlying the detrimental effects of maternal metabolic disorders on offspring health are beginning to be eluci-dated, and many of these may occur through altered epigenetic modifications. Various types of nutrients and metabolites can influ-ence epigenetic modifications[16–18]. For example, vitamin C is essential for proper DNA demethylation and fetal female germline development through modulating TET activity[19]. The biologically active form of vitamin B1 (VB1, thiamin), thiamin pyrophosphate (TPP), is a cofactor of pyruvate dehydrogenase (PDH) in nutrient metabolism, and is

[1]State Key Laboratory of Reproductive Regulation and Breeding of Grassland Livestock (R2BGL), College of Life Sciences, Inner Mongolia University, Hohhot 010070, China. [2]Department of Reproductive Medicine, Qingdao Municipal Hospital, School of Medicine, Qingdao University, Qingdao 266011, China. [3]Department of obstetrical, Qingdao Municipal Hospital, School of Medicine, Qingdao University, Qingdao 266011, China. [4]College of Life Sciences, Institute of Reproductive Sciences, Qingdao Agricultural University, Qingdao 266109, China. ✉e-mail: zhouyang106@126.com; zhangteng428@163.com

required for de novo synthesis of acetyl-CoA and acetylation of core histones[20,21].

Despite extensive findings during investigations into the detrimental effects of maternal high-fat diet (HFD) on metabolic dysfunction, immune, and brain functions of offspring[14,22,23], fewer efforts have focused on the effects of maternal obesity on offspring oogenic fate. The sustained fertility of females throughout reproductive life is determined by the ovarian primordial follicle pool, the assembly of which begins around embryonic day 17.5 (E17.5) in mice[24] and around the 16th week of pregnancy in humans[25]. Thus, the insights needed to focus on the role of non-gamete transmission of mother-to-offspring, influencing offspring oogenic fate determination. Strategies to interrupt this intergenerational transmission can pave the way for halting this harmful cycle.

In this work, we revealed that maternal HFD-induced crosstalk between mother and offspring provided a vital cue, indicating that maternal gut microbiota-related VB1 confers resistance to any deficits during primordial follicle formation in offspring via altering histone acetylation and chromatin accessibility, which promotes mitochondrial function and granulosa cell proliferation.

## Results

### Maternal HFD during pregnancy impaired primordial follicle formation in offspring

To investigate the impact of maternal HFD in pregnancy on ovarian development in the fetus and neonate, pregnant C57BL/6 mice were fed with either a normal diet (mND) or a HFD (mHFD) after mating (Fig. 1a). On days 5.5–19.5 of gestation, mHFD mice became notably overweight (Supplementary Fig. 1a). Intriguingly, the body weight, body length, and ovary size of newborns at postnatal day 0 (P0) and P3 from mHFD mothers were less than those of offspring from mND mothers, which implied fetal growth retardation in mHFD mice (Supplementary Fig. 1b, c). As primordial follicle assembly starts during the fetal stage, the number of unassembled oocytes (germ cells in nests) and primordial follicles (germ cells within follicles) in the offspring from mND and mHFD mice at P0 and P3 were examined by immunostaining with DDX4 antibody (Fig. 1b). Notably, the percentage of primordial follicles was dramatically decreased in offspring from mHFD mice compared to mND mice (Fig. 1c). However, there was no significant difference in the total number of oocytes (Fig. 1d). As expected, some oocyte-specific transcription factors, consistent with the timing of primordial follicle formation, were downregulated in the offspring of mHFD mice (Fig. 1e–g). Moreover, there were fewer primordial follicles and growing follicles in the ovaries of 3-week-old and 7-month-old offspring from mHFD mice (Fig. 1h and Supplementary Fig. 1d), indicating that the defects of ovarian reserves had a long-range effect. In addition, we performed the experiments using ICR mice, and observe identical phenotypes over the same period of HFD feeding (Supplementary Fig. 1e–h). These data confirmed that primordial follicle formation was disrupted from mHFD offspring. In addition to the loss of primordial follicles, oocyte quality and developmental potentiality were also compromised in the ovaries of 3-week-old offspring from mHFD mice (Supplementary Fig. 2). Specifically, the rate of aberrant spindles and misaligned chromosomes was significantly increased in the mHFD group (Supplementary Fig. 2a, b). In addition, the abnormal proportion of 2-cell embryos, 4-cell embryos, and blastocysts was significantly increased after in vitro fertilization in the mHFD group (Supplementary Fig. 2c, d). These findings suggest that the maternal HFD during pregnancy had long-lasting effects on the ovarian primordial follicle reserve and oocyte quality in offspring.

### Maternal VB1 is a dominant factor determining primordial follicle formation in offspring

Fetal nutrient acquisition is dependent on maternal-embryonic crosstalk, in which maternal metabolites play a key role[6]. Therefore, we reasoned that the serum circulating metabolites from HFD mothers may have induced the observed impairment of primordial follicle formation. To test this hypothesis, we examined the untargeted metabolome profiles generated from maternal serum samples using ultra-high performance liquid chromatography-quadrupole time-of-flight mass spectrometry (UPLC-QTOF-MS). Distinct clustering of metabolites was apparent between mND and mHFD groups according to principal components analysis (PCA) and orthogonal partial least squares-discrimination analysis (OPLS-DA) (Fig. 2a and Supplementary Fig. 3a). The abundance of 454 metabolites were significantly altered between mND and mHFD groups (Supplementary Fig. 3b). In addition, the pathway of vitamin digestion and absorption was remarkably prominent (Fig. 2b), and six metabolites within this pathway showed dramatic changes in the mHFD group (Fig. 2c). More importantly, the abundance of VB1 was highly correlated with offspring phenotype, especially the percentage of the germline in nests or follicles (Supplementary Fig. 3c, d). Accordingly, the levels of VB1 in maternal serum and offspring's P3 ovaries were significantly decreased in the mHFD group compared with the mND group (Fig. 2d).

To investigate the potential function of VB1 in primordial follicle formation, we constructed VB1 deficiency and supplementation models as shown in Fig. 2e and Supplementary Fig. 4a (Refer to the Methods for a detailed description of the model). At gestation day 19.5, both the mTA and mTD mice displayed no significant changes in body weight compared to the mND mice (Supplementary Fig. 4b, c). VB1 supplementation did not exert a significant effect on the body weight of pregnant individuals in the mHFD mice (Supplementary Fig. 4b, c). In addition, no statistically significant disparities in litter size were observed across the experimental cohorts (Supplementary Fig. 4d). Notable, after exposure to 60 mg/kg of the VB1 antagonist, Amprolium (mTA), for 14 days or a VB1 deficiency-diet (mTD) fed for 7 days during pregnancy, the levels of VB1 in maternal serum, the development of primordial follicle formation, and the growth indexes of fetus closely resembled those observed in the pregnant maternal HFD model (Fig. 2f–h and Supplementary Fig. 4e–h). Furthermore, VB1 supplementation during pregnancy promotes maternal serum VB1 levels and primordial follicles formation as well as the growth indexes of fetus (Fig. 2f–h and Supplementary Fig. 4e–h). Collectively, we indicated that that maternal VB1 during pregnancy protected offspring from impaired primordial follicle formation.

The gut microbiota is a pivotal modulator of nutrient metabolism and absorption. In order to determine whether the maternal gut microbiota influences maternal VB1 level, we examined 16 S ribosomal DNA (rDNA) amplicon sequencing (Supplementary Fig. 5a–d). Principal coordinate analysis (PCoA) based on Bray-Curtis confirmed that there was a distinct clustering between mND and mHFD groups (Supplementary Fig. 5e). Furthermore, the 43 bacteria were altered at the genus level in the mHFD group compared with the mND group (Supplementary Fig. 5f). Moreover, we verified that the mHFD group exhibited increased levels of VB1 in intestinal digesta (Supplementary Fig. 5g). Given that the main source of VB1 in the body is from food intake, we hypothesis whether the dysbiosis of micro-ecological balance may disrupt VB1 absorption. Intestinal histopathological results showed that there were no differences between the two groups (Supplementary Fig. 5h). Notably, the drastic reduction of the protein expression level of VB1 transporter SLC19A3 suggested that the absorption of VB1 was disrupted in the jejunum and ileum (Supplementary Fig. 5i).

As shown in the fecal microbiota transplantation (FMT) experiment (Supplementary Fig. 6a), we further investigated the contribution of the maternal gut microbiota to VB1 absorption. As expected, the levels of VB1 were increased, and the protein expression levels of SLC19A3 was decreased in intestinal digesta from mHFD recipient mice (mHFD-FMT) compared with mND recipient mice (mND-FMT), respectively (Supplementary Fig. 6b, c). Notably, significantly decreased levels of VB1 were observed in both maternal serum and

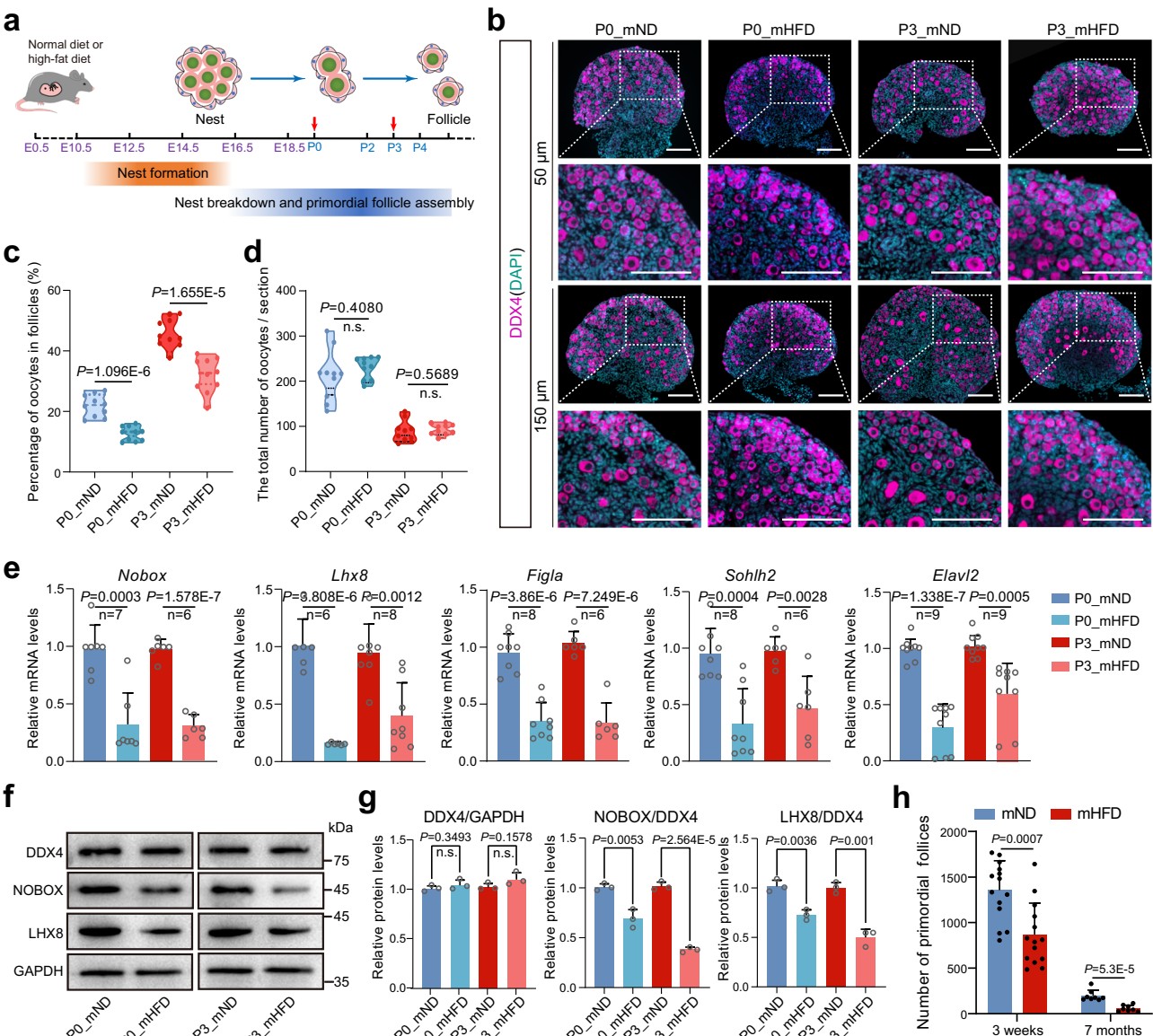

**Fig. 1 | Maternal HFD during pregnancy led to impairment of primordial follicle formation in offspring. a** Schematic illustration of the maternal diet regimen and timeline of primordial follicle formation. The detection time point is indicated by a red arrow. **b** IF staining of DDX4 in offspring ovaries from mND and mHFD groups, and images were captured at two specific locations: 50 µm and 150 µm from the beginning of the tissue edge. DDX4 and DNA are stained in magenta and dark indigo, respectively. DDX4 indicate oocytes. Scale bar, 100 µm. **c** Violin plot showing the percentages of oocytes within follicles (*n* = 10 biologically independent repeats from 10 litters for each group). The upper and lower boundary in the plot indicates the upper and lower quantiles, the line inside the plot the median. **d** Violin plot showing the total number of oocytes per section in ovary in each group (*n* = 10 biologically independent repeats from 10 litters for each group). The upper and lower boundary in the plot indicates the upper and lower quantiles, the line inside the plot the median. **e** RT-qPCR analyzing the expression of *Nobox*, *Lhx8*, *Sohlh2*, *Figla* and *Elavl2* in each group. The number of biologically independent repeats is indicated (*n*). **f, g** Representative images and relative protein levels of DDX4, NOBOX and LHX8 in offspring ovaries from mND and mHFD groups. DDX4 and GAPDH were loading controls (*n* = 3 biologically independent repeats). Uncropped blots in Source Data. **h** Number of primordial follicles at 3 weeks (*n* = 14 biologically independent repeats) and 7 months (*n* = 8 biologically independent repeats) in offspring ovaries from mND and mHFD groups. Data were all presented as mean ± SD. A Student's *t* test (two-tailed) was used for statistical analysis (**c**–**e**, **g**, **h**); n.s., not significant. Source data are provided as a Source Data file.

offspring ovaries in the mHFD-FMT group compared to the mND-FMT group (Supplementary Fig. 6d, e). These findings suggest that maternal HFD disrupts the balance of gut microbiota, leading to impaired absorption of VB1 during pregnancy.

## Maternal HFD during pregnancy induced divergent transcriptional profiling of primordial follicle formation in ovaries of offspring

To characterize the mechanistic features of maternal HFD-induced impaired primordial follicle assembly, a 10× Genomics single-cell atlas of ovarian follicle development at P0 and P3 was performed (Fig. 3a).

We filtered low-quality cells based on the number of genes per cell and the count of each gene (Supplementary Fig. 7a, b). The major cell populations from the mND and mHFD groups were clustered by using uniform manifold approximation and projection (UMAP), and 12 cell clusters were identified on the basis of their distinct transcriptomic signatures (Supplementary Fig. 7c). Corresponding to known clustering of different cell types from the neonatal ovary[26], we identified six major cell types including germ cells (clusters 4, 6) with *Ddx4* and *Dazl* expression, granulosa cells (clusters 0, 2, 3, 10) with *Amhr2* and *Kitl* expression, stromal cells (clusters 1, 9, 11) with *Mfap4* and *Col1a1* expression, endothelial cells (cluster 8) with *Egf17* and *Aplnr*

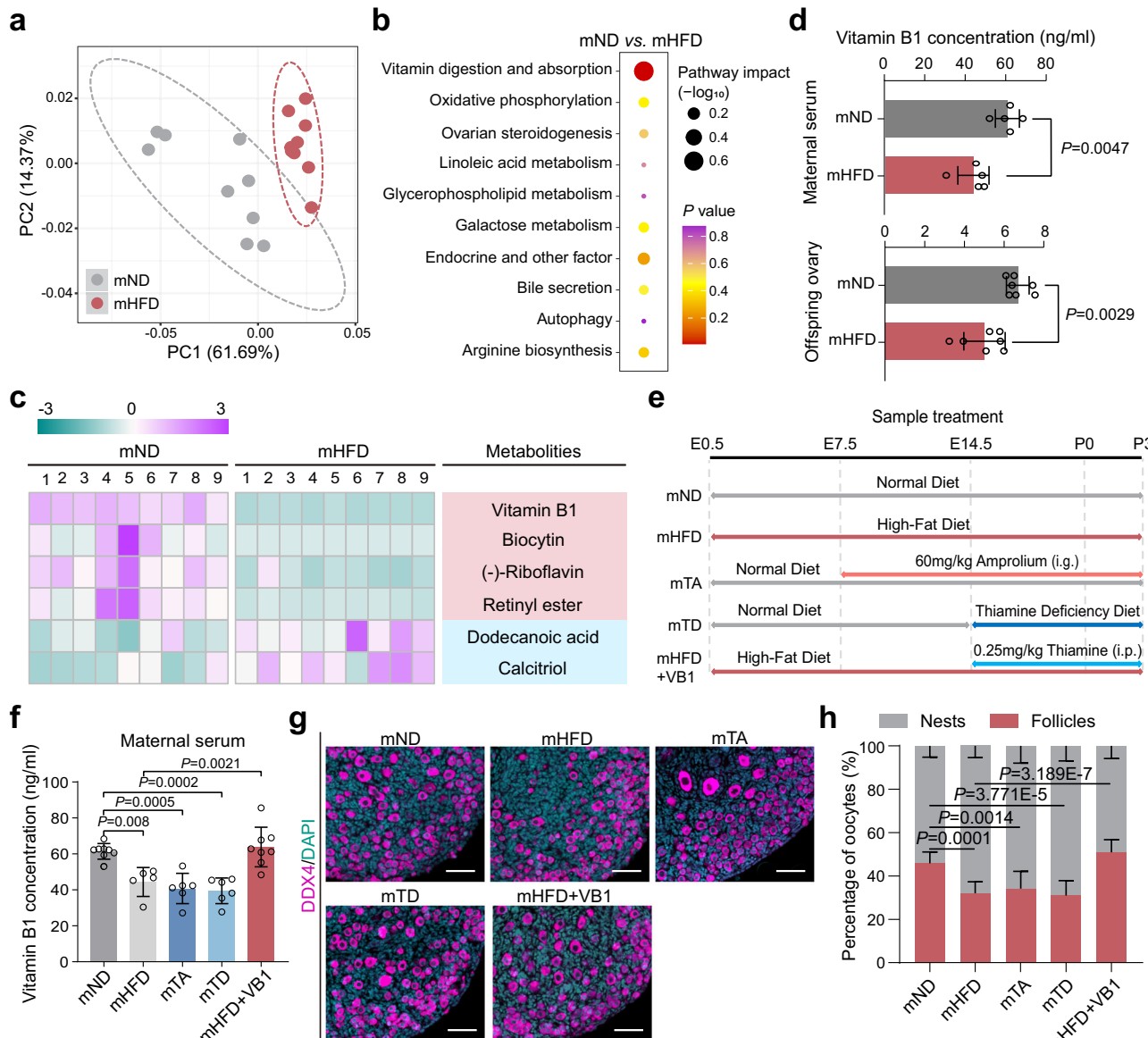

**Fig. 2 | Depletion of the maternal vitamin B1 impairs primordial follicle assembly in offspring. a** Principal component analysis score plot for discriminating the maternal serum metabolome from the mND and mHFD groups ($n = 9$ mice for each group). **b** Disturbed metabolic pathways in the mND *vs.* mHFD groups. The *P*-value is obtained through hypergeometric distribution (two-tailed) and subsequently adjusted using False Discovery Rate (FDR) correction. **c** Heatmaps of the differential metabolites in vitamin digestion and absorption pathways. **d** The vitamin B1 concentration in maternal serum ($n = 5$ biologically independent repeats) or in offspring ovary ($n = 7$ biologically independent repeats) in each group on P3. **e** Study design of sample treatment experiment. **f** The vitamin B1 concentration in maternal serum in the mND ($n = 8$ biologically independent

repeats), mHFD ($n = 5$ biologically independent repeats), mTA ($n = 6$ biologically independent repeats), mTD ($n = 6$ biologically independent repeats), and mHFD +VB1 ($n = 8$ biologically independent repeats) on P3. **g** IF staining of DDX4 in offspring ovaries in the indicated groups on P3. DDX4 and DNA are stained in magenta and dark indigo, respectively. DDX4 indicate oocytes. Scale bar, 50 µm. **h** The percentages of germ cells within nests and follicles in mND, mHFD, mTA, mTD and mHFD+VB1 groups, respectively ($n = 10$ biologically independent repeats from 10 litters for each group). Data were all presented as mean ± SD. In **d**, the two-tailed student's t test was used for statistical analysis; In (**f**, **h**) statistical analyses were performed by one-way analysis of variance (ANOVA) with Tukey's test for multiple comparisons. Source data are provided as a Source Data file.

expression, erythrocytes (cluster 5) with *Alas2* and *Rhd* expression, and immune cells (cluster 7) with *Cd52* and *Cd53* expression (Fig. 3b and Supplementary Fig. 7d).

Subsequently, we extracted germ cells subpopulation and monocle were performed in-depth analysis using R Package "Monocle" (Supplementary Fig. 7e, f). As shown in Supplementary Fig. 7g, branched point from three states of germ cells were observed. Notably, we compared cell occupancy in different cell states, and found that the rate of state 2 was almost lost in mHFD groups (Fig. 3c), which suggested that state 2 may play pivotal role in the development of primordial follicle formation (Fig. 3c). To

explore the genetic characteristics of germ cells at state 2, Gene Ontology (GO) enrichment analysis was performed to investigate the gene function categories (Fig. 3d). Obviously, the GO terms of 'mitochondrion organization', 'mitochondrial respiratory chain complex assembly', and 'ATP metabolic process' were highly enriched in germ cells at state 2 (Fig. 3d), and suggested that mitochondrial morphology and function were disrupted in mHFD groups during primordial follicle assembly. Accordingly, the expression of representative transcripts of mitochondrial organization were validated by RT-qPCR (Fig. 3e). In addition, the mitochondrial morphology of germ cells from the mND and mHFD

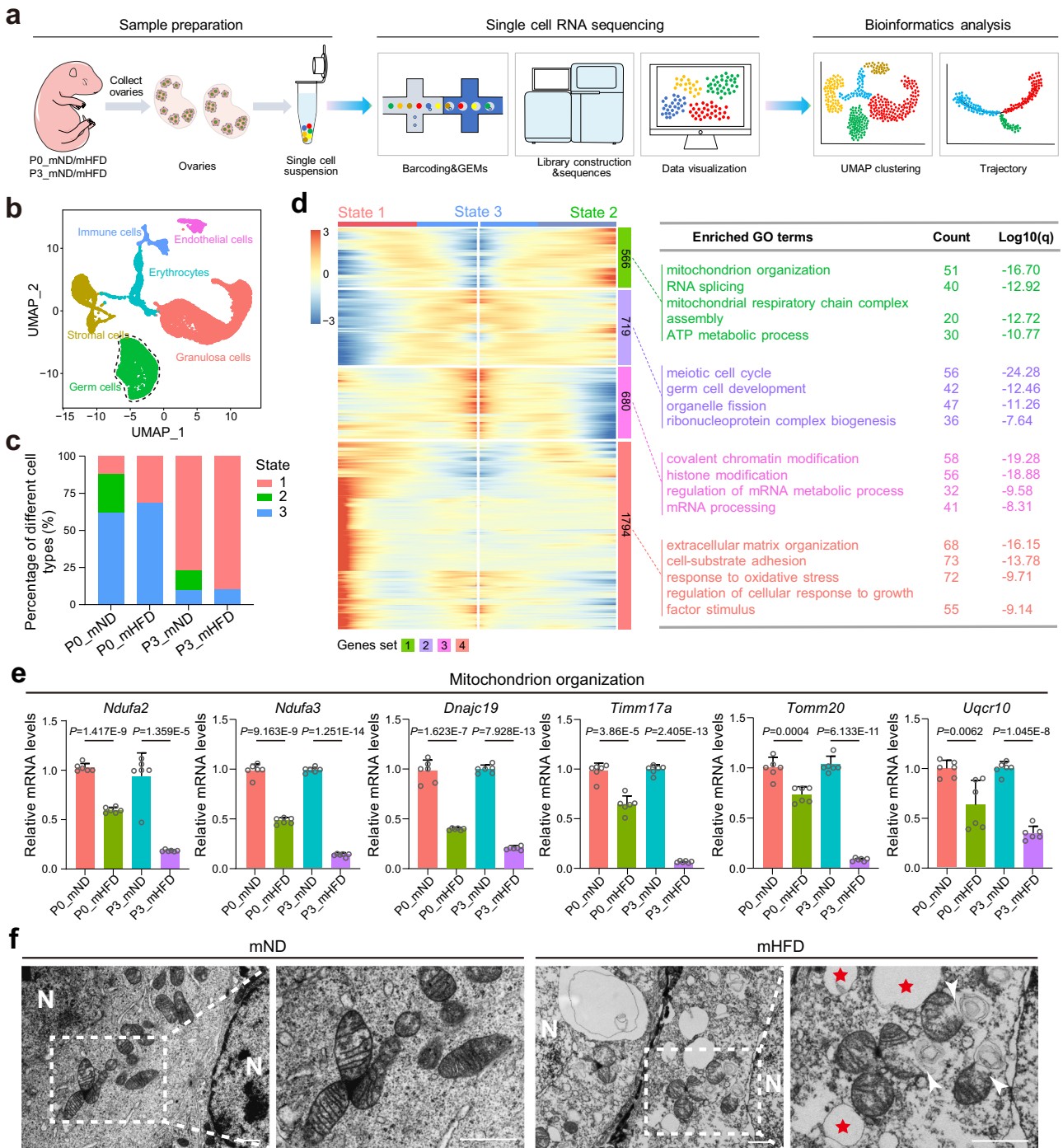

**Fig. 3 | Maternal HFD during pregnancy induced mitochondrial dysfunction of germ cells. a** Schematic diagram of the scRNA-sequencing analysis procedure. **b** UMAP plot of six main ovarian cell types. **c** Percentages of three cell states. **d** Heatmap representing the expression of four gene sets with three cell states (left); GO terms of differentially expressed genes in each gene set (right). **e**, RT-qPCR analyzing the expression of mitochondrial organization related-genes in offspring ovaries from mND and mHFD groups ($n = 6$ biologically independent repeats). **f** Transmission electron microscopic imaging of mitochondria in germ cell at P3. Red asterisks indicate lipid droplet, white arrows indicate abnormal mitochondria and white N indicate germ cell nuclei. Scale bar, 1 μm. Data were all presented as mean ± SD. A Student's *t* test (two-tailed) was used for statistical analysis (**e**). Source data are provided as a Source Data file.

groups was observed by transmission electron microscopy. Ultrastructural aberrations of mitochondria were frequently observed in the neonatal ovaries of the offspring from the mHFD mice, which were characterized by vacuolation, cristae fragmentation, and mitochondrial membrane rupture (Fig. 3f). Taken together, these results indicated that maternal HFD induced mitochondrial dysfunction of germ cells during the development of primordial follicle formation in offspring.

## Maternal VB1 participation in acetyl-CoA metabolism is necessary for primordial follicle assembly in offspring

Thiamin pyrophosphate (TPP), an active form of VB1, is a coenzyme of pyruvate dehydrogenase complex E1 (PDC-E1 or PDH), and is responsible for the conversion of pyruvate to acetyl-CoA (Fig. 4a)[20,21]. Hence, we reasoned whether VB1 insufficiency would lead to substantially reduced PDH activity and acetyl-CoA level. As expected, sharp declines in PDH activity and acetyl-CoA levels were detected in the P3 ovaries of

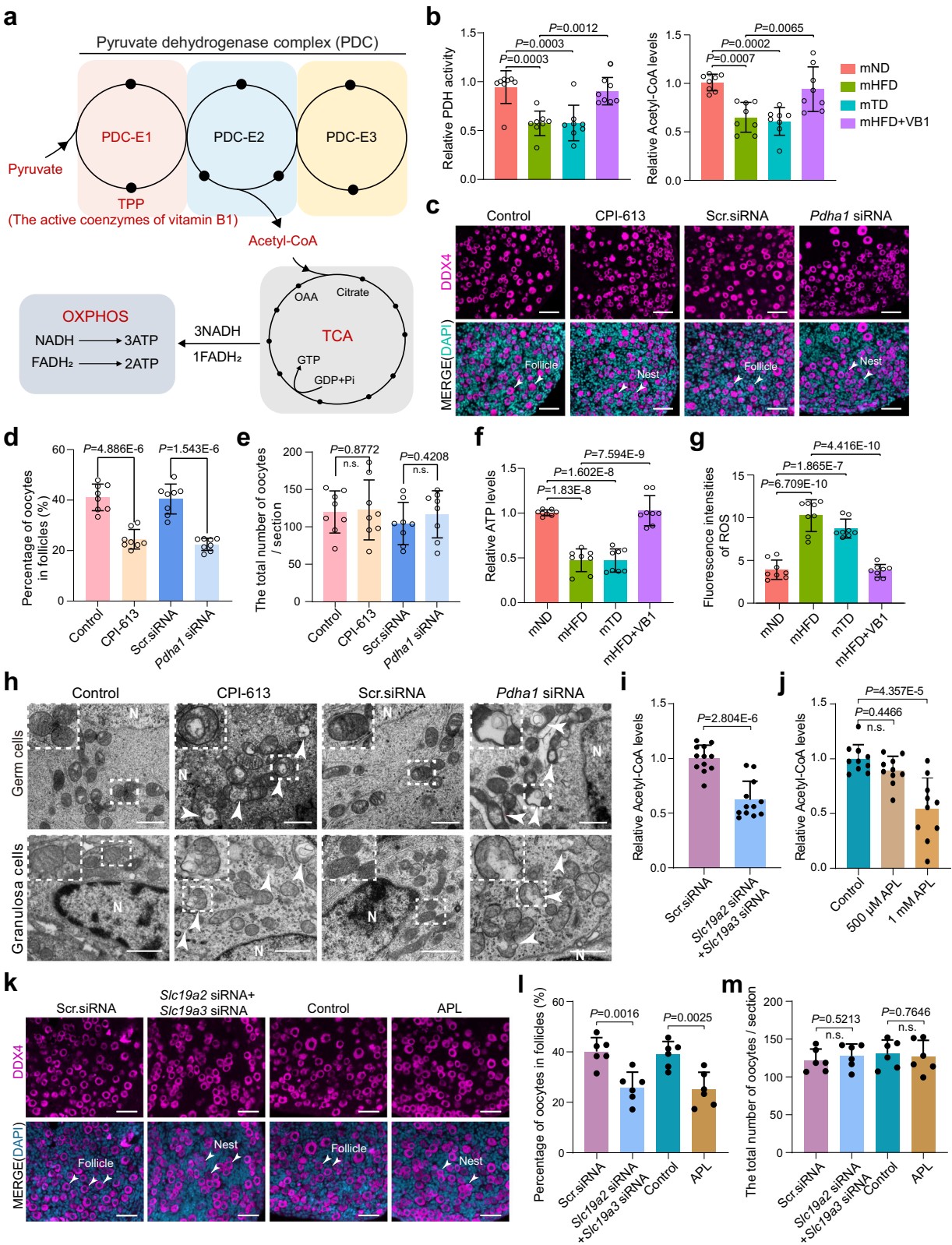

offspring from mHFD and mTD mice (Fig. 4b). In contrast, PDH activity and acetyl-CoA level were remarkably elevated by VB1 supplementation in the P3 ovaries of offspring from mHFD mice (Fig. 4b), indicating that VB1 is indispensable in the process of synthesizing acetyl-CoA from pyruvate in the ovaries of fetuses and neonates. In addition, using an in vitro ovarian culture system, the optimal transfection conditions were created, and siRNA-678 was selected as

the suitable interfering siRNA for *Pdha1* (Pyruvate Dehydrogenase E1 Subunit Alpha 1) (Supplementary Fig. 8a). It was demonstrated that the *Pdha1* siRNA and CPI-613 groups closely resembled the maternal HFD-induced defects of primordial follicle assembly (Fig. 4c−e and Supplementary Fig. 8b−d). Additionally, a significant decrease in PDH activity and acetyl-CoA levels were observed in the *Pdha1* siRNA-treated ovaries or PDH inhibitor, CPI-613-treated ovaries compared to

**Fig. 4 | Maternal vitamin B1 is crucial for mitochondrial function and primordial follicle in neonatal ovary. a** Schematic illustration of vitamin B1 participation in acetyl-CoA metabolism. **b** Relative pyruvate dehydrogenase (PDH) activity and relative Acetyl-CoA levels in offspring ovaries from mND, mHFD, mTD or mHFD +VB1 mice, respectively ($n = 8$ biologically independent repeats). **c** IF staining of DDX4 in ovaries treated with PDH inhibitor CPI-613, Scr-siRNA or *Pdha1*-siRNA, respectively. DDX4 and DNA are stained in magenta and dark indigo, respectively. DDX4 indicate oocytes. Scale bar, 50 μm. **d** The percentages of oocytes within follicles in the indicated groups ($n = 8$ biologically independent repeats). **e** The total number of oocytes per section in the indicated groups ($n = 8$ biologically independent repeats). **f, g** Relative ATP levels and fluorescence intensity of ROS in offspring ovaries from mND, mHFD, mTD or mHFD+VB1 mice, respectively ($n = 8$ biologically independent repeats). **h** Transmission electron microscopy imaging of mitochondria in germ cells and granulosa cells in the indicated groups ($n = 3$ biologically independent repeats). White arrows indicate abnormal mitochondria and white N indicate germ cell nuclei. Scale bar, 1 μm. **i** Relative Acetyl-CoA levels of Scr.siRNA-treated or *Slc19a2*-siRNA+*Slc19a3*-siRNA-treated ovaries ($n = 12$ biologically independent repeats). **j** Relative Acetyl-CoA levels of Control-treated, 500 μM or 1 mM amprolium (APL)-treated ovaries ($n = 10$ biologically independent repeats). **k** IF staining of DDX4 in ovaries treated with 1 mM APL, Scr-siRNA or *Slc19a2*-siRNA +*Slc19a3*-siRNA, respectively. DDX4 and DNA are stained in magenta and dark indigo. Scale bar, 50 μm. **l** The percentages of oocytes within follicles in ovaries treated with 1 mM APL, Scr-siRNA or *Slc19a2*-siRNA+*Slc19a3*-siRNA, respectively ($n = 6$ biologically independent repeats). **m** The total number of oocytes per section in ovaries treated with 1 mM APL, Scr-siRNA or *Slc19a2*-siRNA+*Slc19a3*-siRNA, respectively ($n = 6$ biologically independent repeats). Data were all presented as mean ± SD. Statistical analyses were performed by one-way analysis of variance (ANOVA) with Tukey's test for multiple comparisons (**b, f, g, j**) or two-tailed student's *t* test (**d, e, i, l, m**); n.s. not significant. Source data are provided as a Source Data file.

control-treated ovaries (Supplementary Fig. 8e, f), which suggested that PDH involvement in the conversion of pyruvate to acetyl-CoA is essential during primordial follicle assembly.

Given that maternal HFD induced mitochondrial dysfunction in neonatal ovaries. Meanwhile, acetyl-CoA is required for mitochondrial TCA cycles, we measured the ATP content in P3 ovaries. ATP levels were notably declined in *Pdha1* siRNA or CPI-613-treated ovaries, as in the mHFD and mTD ovaries, but levels recovered following VB1 supplementation (Fig. 4f and Supplementary Fig. 8g), revealing that mitochondrial function might be compromised by reduced acetyl-CoA level owing to VB1 insufficiency. It is known that mitochondrial dysfunction causes ROS generation and its ultrastructural aberrations. Much stronger ROS signals appeared in the ovaries *of Pdha1* siRNA or CPI-613-treated groups than in control groups, which was consistent with the mHFD and mTD groups (Fig. 4g and Supplementary Fig. 8h–j). Meanwhile, ultrastructural aberrations of mitochondria were frequently observed in germ cells and granulosa cells in the ovaries from *Pdha1* siRNA or CPI-613-treated groups (Fig. 4h); this was coordinated with the discrepant transcripts of mitochondrion-related by 10×genomics scRNA-sequencing.

Furthermore, we investigated the impact of VB1 deficiency on the ovaries through blocking two vitamin B1 transport proteins using in vitro culture and RNA interference (RNAi) targeting *Slc19a2* and *Slc19a3*, along with Amprolium (APL) to inhibit VB1 utilization[27,28]. The results demonstrated that *Slc19a2* siRNA-1547 and *Slc19a3* siRNA-236 effectively inhibited the protein expression of SLC19A2 and SLC19A3, respectively (Supplementary Fig. 9a, b). Interestingly, inhibiting SLC19A2 led to an upregulation of SLC19A3, while inhibiting SLC19A3 did not affect SLC19A2 expression (Supplementary Fig. 9c, d), indicating a compensatory mechanism between these transport proteins. Co-transfection of *Slc19a2* and *Slc19a3* siRNAs effectively reduced SLC19A2/SLC19A3 expression and resulted in a significant decrease in ovarian tissue acetyl-CoA content (Fig. 4i and Supplementary Fig. 9e, f). In addition, we employed 500 μM and 1 mM APL in vitro to inhibit vitamin B1 utilization, resulting in a significant decrease in acetyl-CoA content at 1 mM APL (Fig. 4j). Crucially, both the *Slc19a2*/*Slc19a3* siRNA group and the 1 mM APL group exhibited significant inhibition of primordial follicle formation compared to their respective control groups, while the total number of follicles remained unchanged (Fig. 4k–m). These results closely resembled the pregnant maternal HFD model, illustrating that normal conversion of pyruvate to acetyl-CoA, mediated by VB1, is vital for primordial follicle assembly in the neonatal ovary.

## Maternal VB1 is crucial for the acetylation modification of histones in neonatal ovaries
Consistent with previous studies[21], the subunits of the PDC-E1 were also present in the nuclei of ovarian cells (Fig. 5a–c). Given that nuclear acetyl-CoA is required for histone acetylation and epigenetic regulation (Fig. 5d), the levels of core histone acetylation, such as Ac-H3, Ac-H4, Ac-H3K9, and Ac-H3K18, were assessed among each group of P3 ovaries. Histone acetylation levels were reduced notably in the ovaries of offspring from mHFD and mTD mice (Fig. 5e, f). Furthermore, VB1 supplementation remarkably recovered the histone acetylation levels in ovaries of offspring from mHFD mice (Fig. 5e, f). Additionally, the detection of histone acetylation levels revealed that both the *Slc19a2*/*Slc19a3* siRNA group and the APL group exhibited significant reductions in Ac-H3 and Ac-H4 levels compared to the control group (Fig. 5g, h). Similar phenotypes were also exhibited in *Pdha1* siRNA-treated or CPI-613-treated ovaries (Supplementary Fig. 10). These data indicated that the conversion of pyruvate to acetyl-CoA via PDH, especially VB1, is involved in the process of nuclear histone acetylation in neonatal ovaries. In summary, VB1 participated in the metabolism of pyruvate to acetyl-CoA mediated by PDH, which is crucial for the acetylation modification of histones and mitochondrial function involved in primordial follicle assembly in the neonatal ovary.

## Maternal VB1 insufficiency influenced granulosa cell proliferation modulated by chromatin accessibility
Histone acetylation is closely related to gene expression through the modulation of chromatin accessibility. Thus, ATAC-sequencing was employed to identify the status of global chromatin accessibility in P3 ovaries of offspring from the mND and mHFD groups. By analyzing the ATAC-sequencing data, we identified that the ATAC signals around the transcription start site (TSS) of the entire genome were decreased in the mHFD group as compared to the mND group (Fig. 6a). Specifically, the mHFD group showed significantly downregulated differentially accessible regions (DARs), which were predominantly located in the promoter regions (Fig. 6b, c). These genes at the promoter regions with reduced accessibility were predominantly enriched in the 'Protein processing in endoplasmic reticulum', 'Ribosome', and 'Cell cycle' pathways (Fig. 6d). Moreover, significant differentially expressed genes (DEGs) in 16,505 downregulated genes, as compared to the 880 upregulated genes were shown in the granulosa cells of ovaries from offspring at P0 and P3 by using 10× genomics scRNA-sequencing analysis (Fig. 6e and Supplementary Fig. 11a, b); this was consistent with the data shown in Fig. 6b. Common downregulated genes of granulosa cells between P0 and P3 were significantly enriched in the 'Cell cycle' pathway (Fig. 6f and Supplementary Fig. 11c). By analyzing ATAC-sequencing and 10×genomics scRNA-sequencing, the overlapped genes enriched in 'Cell cycle' pathway were identified (Fig. 6g and Supplementary Fig. 11d), including *Cdk4*, *Cdk7*, *Mcm6*, *Orc3*, *Ccnh*, *Anapc16*, *Atm*, *Smc3*, *Pttg1*, and *Cdkn1b*. Among them, *Cdk4*, *Cdk7*, *Mcm6*, *Orc3*, and *Ccnh* were involved in the 'G1-S phase progression and DNA biosynthesis' processes. We assessed the changes in ATAC-sequencing peaks within these gene loci qualitatively, and found that the boxed peaks in mHFD groups were significantly reduced compared with mND groups (Fig. 6h and Supplementary Fig. 11e). Additionally,

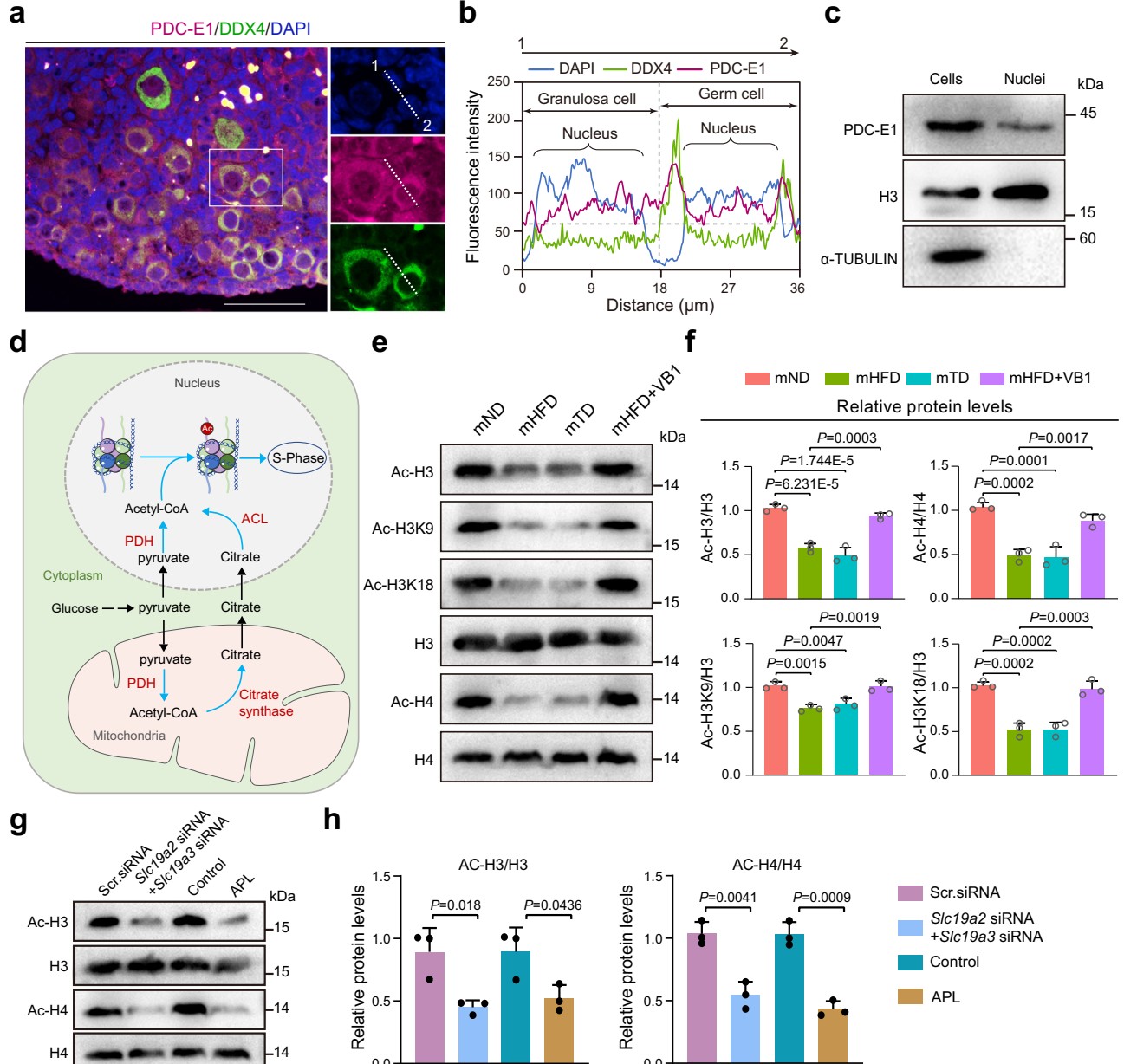

**Fig. 5 | Maternal vitamin B1 is crucial for the modification of histones acetylation in neonatal ovary. a** IF staining of PDC-E1 and DDX4 in ovary. PDC-E1, DDX4 and DAPI are stained in magenta, green and blue, respectively (*n* = 4 biologically independent repeats). Scale bar, 50 µm. **b** Quantify the fluorescence intensity of PDC-E1, DDX4, and DAPI in granulose and germ cells along the white dashed lines (**a**). The horizontal dashed gray lines (**b**) indicate the baseline fluorescence intensity, while the vertical dashed gray lines represent the demarcation between germ cells and granulosa cells. **c** The relative protein levels of PDC-E1 in ovarian cells and nuclei (*n* = 3 biologically independent repeats). H3 and α-TUBULIN were loading controls. Uncropped blots in Source Data. **d** Schematic illustration of the conversion of pyruvate to acetyl-CoA via PDH in nucleus and mitochondria.

**e**, **f** Representative image and relative protein levels of Ac-H3, Ac-H3K9, Ac-H3K18 and Ac-H4 in offspring ovaries from mND, mHFD, mTD or mHFD+VB1 mice, respectively. H3 and H4 were loading controls (*n* = 3 biologically independent repeats). Uncropped blots in Source Data. **g**, **h** Representative image and relative protein levels of Ac-H3 and Ac-H4 in ovaries treated with 1 mM APL, Scr-siRNA or *Slc19a2*-siRNA+*Slc19a3*-siRNA, respectively. H3 and H4 were loading controls (*n* = 3 biologically independent repeats). Uncropped blots in Source Data. Data were all presented as mean ± SD. Statistical analyses were performed by one-way analysis of variance (ANOVA) with Tukey's test for multiple comparisons (**f**) or two-tailed student's *t* test (**h**). Source data are provided as a Source Data file.

significantly lower levels of peaks and gene expression were also noted in 'Oxidative phosphorylation' and 'TCA cycle' processes from common genes of both ATAC-sequencing and scRNA-sequencing (Supplementary Fig. 11d, f, g). Furthermore, the protein levels of these genes were dramatically decreased in the ovaries of offspring from mHFD and mTD mice, which is consistent with *Pdha1* siRNA or CPI-613-treated ovaries (Fig. 6i, j and Supplementary Fig. 12a, b). VB1 supplementation remarkably recovered these gene levels in the ovaries of offspring from mHFD mice (Fig. 6i, j). These findings

indicated that reduced accessibility in the promoter regions primarily contributed to a marked reduction of 'Cell cycle'-related genes expression levels, which are associated with diminished chromatin accessibility in ovarian granulosa cells of offspring from mHFD group.

The next key question is whether the compromised cell cycle progression was correlated to primordial follicle assembly. The distribution of G0/G1 phase were dramatically increased in granulosa cells treated with *Pdha1*-siRNA or CPI-613 in vitro, which indicated cell cycle

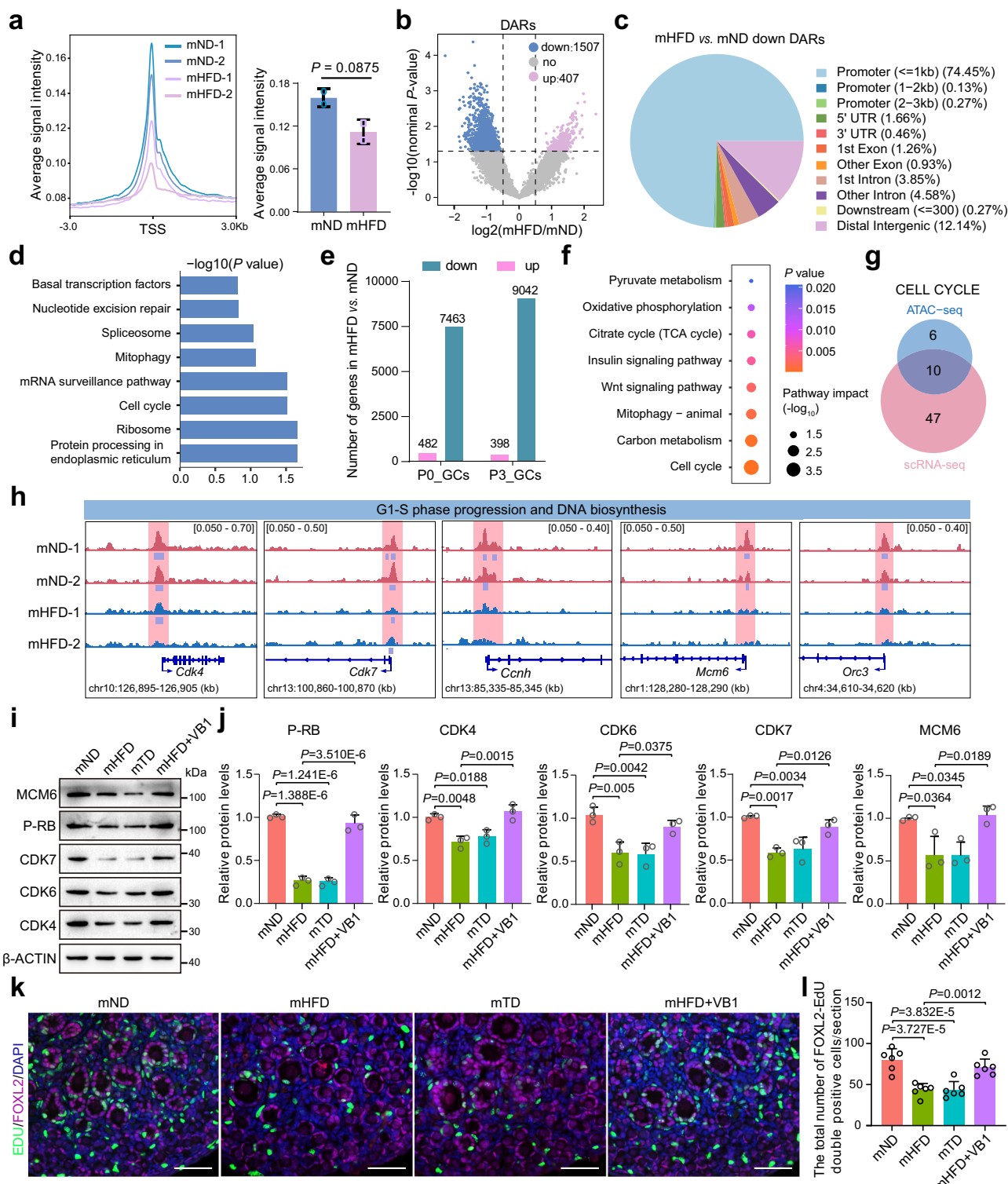

insufficiency not only suppressed both acetyl-CoA and histone acetylation levels, but also had an important impact on cell cycle related-gene expression modulated by chromatin accessibility in ovarian granulosa cells, which led to disruption of primordial follicle assembly.

arrest (Supplementary Fig. 12c, d). Given that the proliferation of granulosa cells which migrate to surround oocytes is indispensable in the formation of primordial follicles, EdU staining was performed to assess granulosa cell proliferation. The data showed that little EdU staining was noted in granulosa cells from the mHFD and mTD groups, which was similar to the *Pdha1*-siRNA or CPI-613 groups, suggesting that aberrant proliferation of granulosa cells was primarily responsible for the inadequate number of primordial follicles (Fig. 6k, l and Supplementary Fig. 12e, f). Notable, VB1 supplementation recovered the proliferation of granulosa cells (Fig. 6k, l). Cumulatively, VB1

### Maternal VB1 levels were associated with gestational diabetes mellitus

Gestational diabetes mellitus (GDM) is a pregnancy-related disorder characterized by high blood sugar levels, and its underlying mechanisms and optimal management strategies remain unknown. The 62

**Fig. 6 | Maternal vitamin B1 insufficiency suppressed granulosa cells proliferation in offspring. a** ATAC-sequencing signals spanning the entire genome visualized in a TSS-centric manner (*n* = 2 biologically independent repeats). **b** Volcano plot of the differentially accessible regions (DARs) between mND and mHFD groups. **c** Genomic distribution of downregulated DARs in the mHFD *vs.* mND groups. **d** KEGG enrichment analysis of genes along with reduced accessibility in the promoter regions. The *P*-value is obtained through hypergeometric distribution (two-tailed) and subsequently adjusted using FDR correction. **e** The number of differentially expressed genes in the granulosa cells of offspring ovaries from mND and mHFD groups. **f** KEGG enrichment analysis of downregulated genes in the granulosa cells of offspring ovaries from mND and mHFD groups. The *P*-value is obtained through hypergeometric distribution (two-tailed) and subsequently adjusted using FDR correction. **g** Venn diagram illustrating the relationship of the cell cycle-related genes between ATAC-sequencing and scRNA-sequencing.

**h** ATAC-sequencing normalized reads shown for G1-S phase progression and DNA biosynthesis genes. **i**, **j** Representative images and relative protein levels of MCM6, P-RB, CDK7, CDK4 and CDK6 in offspring ovaries from mND, mHFD, mTD or mHFD +VB1 mice, respectively (*n* = 3 biologically independent repeats). β-ACTIN were loading controls. Uncropped blots in Source Data. **k**, **l** IF staining of EdU (Proliferation labeling) and FOXL2 (Granular cell marker), and quantification of FOXL2-EdU double-positive cells per section in offspring ovaries from mND, mHFD, mTD or mHFD+VB1 mice, respectively (*n* = 6 biologically independent repeats). EdU, FOXL2, and DNA are stained in green, magenta, and blue, respectively. Scale bar, 50 μm. Data were all presented as mean ± SD. Statistical analyses were performed by one-way analysis of variance (ANOVA) with Tukey's test for multiple comparisons (**j**, **l**) or two-tailed student's *t* test (**a**, **b**). Source data are provided as a Source Data file.

pregnant women included as participants were eligible for follow-up. Women of child-bearing age were divided into two cohorts, including healthy controls (HC) and GDM (Supplementary Table 1). We confirmed that women with GDM experienced a marked elevation of glucose (GLU) and triglycerides (TG) levels, as well as glucose intolerance (Fig. 7a–c). To test whether the maternal HFD model aligned with the phenotypic similarities with GDM, we revealed that the hepatocytes containing lipid droplets in the mHFD mice, as demonstrated by H&E staining (Fig. 7d), and GLU and TG levels were significantly elevated in the mHFD group compared with mND mice (Fig. 7e). Concomitantly, HFD-induced glucose intolerance and insulin resistance were significantly accelerated in pregnant mHFD mice, indicating impaired glucose tolerance and insulin sensitivity (Fig. 7f–i).

To provide further evidence of the importance of maternal VB1 in GDM, we examined VB1 levels of serum in HC and GDM individuals. Intriguingly, women with GDM showed a marked reduction of serum VB1 levels (Fig. 7j), which is consistent with the maternal HFD model. More importantly, the abundance of VB1 was highly correlated with GDM phenotype, especially the glucose intolerance (Fig. 7k, l). These observations highlighted the importance of VB1, which protects offspring primordial follicle formation against impairments, indicating that VB1 would be an effective therapeutic candidate for the treatment of GDM.

## Discussion

Increasing evidence indicates that maternal high-fat diet is a major determinant of long-term health in offspring[29,30]. It is now clear that maternal HFD during pregnancy influences offspring health through non-gamete transmission from mother-to-offspring. Here, the discovery of maternal high fat intake during pregnancy inducing VB1 insufficiency provides a central mechanism for a lower number of primordial follicles in offspring. Furthermore, VB1 supplementation of the HFD-fed dams improved the deficits of the ovarian primordial follicle pools in offspring. These results indicate that maternal-embryonic cross-talk mediated by metabolites is crucial for offspring health, which is different from the effects of gamete inheritance on offspring[31,32]. Recent studies also report that maternal exercise improves the metabolic health of offspring via cross-talk between maternal metabolites and the fetus[6,33]. It is known that high-fat diet is intricately linked to host metabolomes and imbalance of the gut microbial ecosystem[34,35]. Previous studies have indicated that a longitudinal analysis of human across 4 years characterizes the microbial effects on host health are mediated by plasma metabolites[36]. For example, 38 microbial are reported to be associated with plasma VB1 levels and may influence cardiometabolic health[36]. Based on our findings, we propose that the diminished absorption of VB1 and the decreased expression of SLC19A3, its associated transport protein, mostly owing to the imbalanced gut microbiota. Evidence that strongly supports this idea was demonstrated by 16 S rDNA sequencing. We found that the abundance of gut microbiota was dramatically altered at the genus level in HFD-fed pregnant mice. Importantly, FMT

experiment provided additional confirmation of the causal link between dysregulation of maternal gut microbiota and the impaired absorption of VB1. Moreover, further studies are needed to explore the mechanisms by which gut microbiota affects the expression of vitamin B1 transport protein.

The gut microbiota produces a variety of metabolites that circulate in the host body, and the input from these metabolites to chromatin are modifiers of the epigenome[17]. Moreover, previous studies in *C.elegans*, *Drosophila* and mammals suggest that disruptions in parental metabolic state lead to alterations in germ cell chromatin status[37,38]. In this study, notably decreased levels of histone acetylation were observed in the ovaries of offspring from mHFD compared with mTD dams. Previous studies show that calorie restriction (CR) can increase global histone acetylation and influence gene expression by changing the levels of ketone body β-hydroxybutyrate (β-OHB), which functions as an important regulator of histone acetylation through inhibiting class I histone deacetylases (HDACs)[39]. The digestion of dietary fiber by gut bacteria produces short-chain fatty acids (SCFAs), including acetate and butyrate, which are additional sources of acetyl-CoA for histone acetylation[40,41]. Butyrate is also an inhibitor of HDACs, which can influence levels of histone acetylation. Unlike β-OHB and butyrate, acetate can be produced de novo from pyruvate metabolism, which supports histone acetylation by acetyl-CoA[42]. In our study, we demonstrated that VB1 was responsible for the conversion of pyruvate to acetyl-CoA via PDH. In addition, VB1 supplementation of the HFD-fed dams promoted the levels of histone acetylation and acetyl-CoA in offspring. It has also been found that levels of acetyl-CoA and histone acetylation are diminished by inhibiting the enzymes ACLY and ACSS2 in an HFD intake model[43].

In mammalian cells, acetyl-CoA is derived primarily from extracellular glucose, which feeds into mitochondrial metabolism and epigenetic modification of the nucleus. Previous studies confirm that nuclear acetyl-CoA synthesis by PDC is required for histone acetylation and gene regulation[21,44]. Similarly, we found that the subunit of PDC-E1 was also present in the nuclei of ovarian cells. It has been widely reported that histone acetylation plays important roles in modulating chromatin accessibility and the expression of specific genes[45,46]. Here, we confirmed that chromatin accessibility was notably downregulated in ovarian granulosa cells of offspring from HFD-fed dams, and these were predominantly located in the promoter regions. Interestingly, the expression levels of proliferation-related genes in granulosa cells were markedly reduced, which is associated with diminished chromatin accessibility. Notably, VB1 supplementation of HFD-fed dams rescued proliferation deficits of ovarian granulosa cells in offspring. These results offer a more in-depth understanding of the causative link and underlying mechanisms of the effects of maternal HFD on primordial follicle assembly in offspring.

It is known that an imbalance in mitochondrial dynamics has been associated with metabolic disease. Previous studies report that maternal HFD-induced metabolic diseases results in mitochondrial

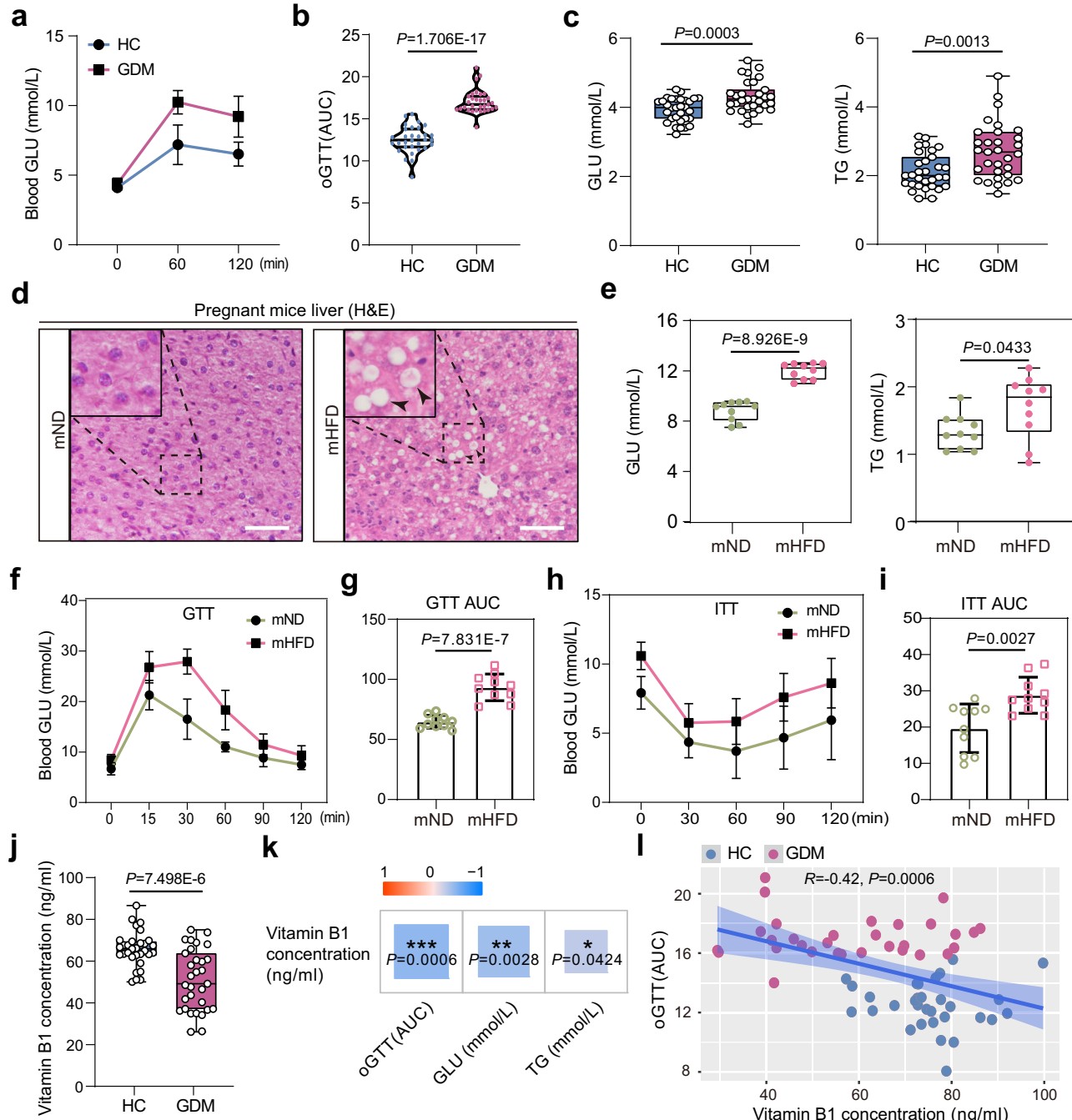

**Fig. 7 | Reduction of vitamin B1 levels in human with gestational diabetes mellitus (GDM). a** Oral glucose-tolerance test (oGTT) for pregnant women with healthy controls (HC) and GDM (*n* = 31 human participants for each group). **b** Violin plot showing area under the curve (AUC) of oGTT for pregnant women with HC and GDM (*n* = 31 human participants for each group). The upper and lower boundary in the plot indicates the upper and lower quantiles, the line inside the plot the median. **c** Box plot showing the levels of glucose (GLU) and triglycerides (TG) in maternal serum of HC and GDM (*n* = 31 human participants for each group). **d** Representative H&E images of liver in mND and mHFD mice. Black arrows indicate lipid droplets. Scale bars, 50 μm. **e** Box plot showing the levels of GLU and TG levels in maternal serum of mND and mHFD groups (*n* = 10 mice for each group). **f, g** Glucose tolerance test (GTT) with AUC in mND and mHFD mice (*n* = 10 mice for each group). **h, i** Insulin tolerance tests (ITT) with AUC in mND and mHFD mice (*n* = 10 mice for

each group). **j** Box plot showing the serum levels of vitamin B1 in maternal serum of HC and GDM (*n* = 31 human participants for each group). **k** Spearman's correlation analysis (two-tailed) between vitamin B1 and oGTT (AUC), and GLU, and TG in HC and GDM, respectively. Blue indicates a negative correlation (*0.01 < *P* < 0.05, **0.001 < *P* < 0.01, ***P* < 0.001). **l** Spearman's correlation analysis (two-tailed) between vitamin B1 and oGTT (AUC) in HC and GDM. *R*-value less than 0 indicates a negative correlation between two variables and *P*-value less than 0.01 was highly significant difference. Data were all presented as mean ± SD. The box in the box plot (**c, e, j**) indicates the upper and lower quartiles, with the line inside the box indicating the median. The whiskers extending from the box represent the range of data, where the lower whisker reaches the minimum, and the upper whisker extends to the maximum. Student's t test (two-tailed) was used for statistical analysis (**b, c, e–j**). Source data are provided as a Source Data file.

dysfunction, which would be passed through the female germline across three generations[47]. The mitochondrial dynamics couple nutrient state to oocyte development via insulin signaling in *Drosophila*[48]. Furthermore, maternal metabolic stress suppresses germline insulin signaling, causing a premature suppression of mitochondrial function, and heritable effects on progeny metabolism[49]. Notably, maternal exercise during pregnancy improves mitochondrial respiration capacity in offspring[50]. In our study, we demonstrated that VB1 treatment notably attenuated the maternal HFD-induced mitochondrial dysfunction of ovarian cells in offspring. The findings above prompted us to consider that VB1 can be used to ameliorate mitochondrial disease in offspring.

The rising incidence of obesity in women has led to an increase in the incidence of GDM[51]. Here, we found that a pregnant maternal high fat intake model showed similar phenotype to GDM. Intriguingly, VB1 abundance was dramatically decreased in the serum samples of humans with GDM. Consistent with this, previous studies have indicated that plasma VB1 level is significantly lower in women with GDM[52]. Meanwhile, a randomized clinical trial has shown that maternal perinatal thiamine supplementation can prevent infantile beriberi[53]. In addition, VB1 deficiency during in utero development causes intrauterine growth retardation in the progeny[54]. Similarly, we also observed that VB1 supplementation was effective in preventing the maternal HFD induction of low birth weight in offspring, although the precise underlying mechanisms of this protective effect remain unclear. In addition, further exploration is needed as to whether glucose tolerance can be improved by VB1 supplementation in mHFD group. Taken together, these findings offer a research avenue into preventive therapies for maternal high fat intake-induced primary ovarian insufficiency (POI) in female offspring and indicates that VB1 is a potential therapeutic candidate for patients with GDM.

## Methods

### Animal and ethics statement

All animal experiments were approved and performed in line with the Animal Care and Use Committee (ACUC) at Inner Mongolia University (SYXK 2020-0006), and SPF Biotechnology Co., Ltd. provided all of the C57BL/6 and ICR mice (Beijing, China). All animal experiments align with ARRIVE guidelines, and animal welfare and health checks were conducted in accordance with the Animal Welfare Act and ACUC guidelines. With the exception of ICR mice utilized in the Supplementary Fig. 1e–h experiment depicted, C57BL/6 mice were employed in all other mouse experiments conducted within this study. All mice were kept in a room with a constant temperature ($23 \pm 2\,°C$), constant humidity (50%), a 12-h light and 12-h dark cycle (lights: 8 a.m.), and free access to food and water. Mating was scheduled on a regular basis at 5 p.m., the presence of the vaginal plug was labeled embryonic day 0.5 (E0.5), and the mice were typically delivered at E19.5, which was termed postnatal day 0 (P0) at the time. Euthanasia procedures were carried out under deep anesthesia by cervical dislocation. After the presence of the vaginal plug, the pregnant mice were randomly assigned to two groups, and fed with a normal diet (mND group, D12451J, Research Diets, Inc.) and a high-fat diet (mHFD group, D12492, Research Diets, Inc.), respectively.

### Human participants

Women Study with healthy controls (HC) people and gestational diabetes mellitus (GDM) were recruited at Maternity and Child Health Hospital of Qingdao (Qingdao, China). The Medicine Ethics Committee of Qingdao Maternity and Child Health Hospital authorized the study (IRB Number: [2021]074). The participants with diabetes or metabolic illnesses, antibiotic use, alcohol or substance misuse, or chronic conditions needing treatment were all excluded from this study. All of the participants provided their written informed consent. At about

26 weeks of gestation, all participants were screened for GDM by an oral glucose tolerance test (oGTT, 75 g glucose). Exceeding any one of the three thresholds, including fasting plasma glucose (5.1 mmol/L), 1-h post-load glucose (10.0 mmol/L), and 2-h post-load glucose (8.5 mmol/L), was consistent with the GDM diagnosis[55]. A total of 31 HC mothers and 31 GDM mothers were recruited in this study (Supplementary Table 1). The study adhered to all pertinent regulations for human study participants and followed the Declaration of Helsinki criteria.

### Glucose tolerance tests and insulin tolerance tests

Postpartum maternal mice with mND and mHFD group were selected with 10 replicates per group. After a 12-h fast, mice were given an intraperitoneal injection of 2 g/(kg body wight) glucose for the glucose tolerance tests (GTT). After glucose injection at the indicated time points (0, 15, 30, 60, 90, 120 min), the blood glucose concentration was determined. In addition, after a 6-h fast, mice were given an intraperitoneal injection of 0.5 IU/(kg body wight) insulin for the insulin tolerance tests (ITT), and the blood glucose concentration was determined at the indicated time points (0, 30, 60, 90, 120 min). For curve drawing, calculating the area under the curve (AUC), and data analysis, GraphPad Prism 8 analytic software (San Diego, CA, USA) was utilized.

### Serum biochemical parameters

Maternal blood samples were collected from participants following an overnight fast of at least 12-h to standardize their basal metabolic state prior to sample collection. The blood samples were centrifuged at 3000 rpm at 4°C for 15 min to separate serum. A Cobas c-311 Coulter chemistry analyzer was used to examine the serum biochemical parameters. Biochemical kits for glucose (GLU, 04404483190) and triglycerides (TG, 20767107322) were purchased from Roche (Shanghai, China).

### Single cell RNA-sequencing (scRNA-sequencing)

**Single cell sample preparation and scRNA-sequencing.** The ovaries of mND and mHFD were collected from the mice at postnatal day 0 (P0) and 3 (P3), respectively. To get a significant quantity of cells and to ensure repeatability, we collected 5 mice from different mothers in each group. After that, we minced the ovary and digested it with collagenase (2 mg/mL, C5138, Sigma-Aldrich, Shanghai, China) for 7 min at 37 °C to produce single cell suspension. Digestion was then terminated with a medium containing 10% serum, filtered with single-cell filter (BD Falcon, 352340, CA, USA) and washed with PBS for 3 times. Subsequently, cell survival fraction and cell population were quantified to match sequencing criteria. Wang et al.[26] described the methods for preparing and sequencing single cell sample libraries in accordance with the manufacturer's recommendations.

**Clustering analysis with Seurat.** The data (CellRanger software v2.2.0 generates) was pre-processed using Seurat package (version 4.0.4) in R (version 4.0.0), and four groups were performed an integrated analysis via "FindIntegrationAnchors" fuoction[56]. Double cells were examined and removed to discard cells with abnormal gene expression from the data using DoubletFinder package (version 2.0.3)[57]. Subsequently, cell clusters were visualized using "UMAP" technology[58] and the "FindAllMarkers" tool was used to find ovarian cell-specific marker genes.

**Single-cell pseudotime trajectory analysis.** The cell clusters trajectory analysis was created using the R Package "Monocle (version 2.1.6)"[59]. Seurat's highly variable genes were utilized as ordering genes to draw the developmental trajectories map of subcellular groups, and then heatmaps showing dynamic genes at branch points calculated using "BEAM" functions.

**Identification of DEGs of scRNA-sequencing.** Desingle software package[60] was performed differential expression analysis of ovary cells in the mND and mHFD groups. Desingle package can reasonable solution the problem of many zero values in scRNA-*sequencing* data. DEGs is defined as a gene with a Desingle corrected *P*-value less than 0.05 and a |log2 fold change| greater than 0.25.

**Gene set functional enrichment analysis.** Enrichment analysis of GO and KEGG was performed using online tools Metascape (version 3.5) (https://metascape.org)[61] and R package "clusterProfiler (version 3.16.1)". EnrichmentMap was created with the DEGs as input using the g.profiler (https://biit.cs.ut.ee/gprofiler/gost), and the results were shown in Cytoscape (version 3.7.2) with *P*-value less than 0.01.

**Immunofluorescence (IF)**
The ovaries were fixed in 4% PFA for 12-h before being processed into 5 μm thick histological sections[62]. After processed followed methods for paraffin sample, the paraffin sections were incubated with primary antibodies (Supplementary Table 2) for 12-h at 4 °C and secondary antibodies (Supplementary Table 2) for 1-h at 25 °C. DAPI was utilized to stain the nuclei, and a coverslip was utilized to pressure the covers. Images were examined with a Nikon fluorescence microscope (A1, Japan). A primordial follicle was characterized as a solitary oocyte surrounded by granulosa cells, whereas aggregates of multiple oocytes are termed as germ cell "nest" (Pepling, 2006). Continuous sections of an intact ovary are counted in every fifth sections and aggregated, as a biological repeat.

**Immunohistochemistry (IHC)**
After deparaffinization for continuous sections of the ovary at 3 weeks of age as described in the IF step, the following IHC procedures were used: The sections were blocked with 3% BSA and 10% donkey serum in TBS after treated with 3% $H_2O_2$, followed by primary antibody incubation for 12-h at 4 °C and secondary antibodies for 1-h at 25 °C after washed with TBST. Subsequently, the sections were stained with peroxidase substrate and counterstained with hematoxylins, and the primordial follicle were examined and counted under a microscope.

**RNA extraction and real time quantitative PCR (RT-qPCR)**
Fresh ovary samples were collected from each treatment group with at least 6 ovaries for this experiment. As directed by the manufacturer, ovarian total RNA extraction and reverse transcription were implemented using EasyPure® RNA Kit (TransGen Biotech, ER101, Beijing, China) and reverse transcription Kit (TransGen Biotech, AT311). The Light Cycler 480 II apparatus (Roche, Germany) was used to perform RT-qPCR utilizing TB Green Premix (TaKaRa, RR820B, Dalian, China). The $2^{(-\Delta\Delta Ct)}$ method was used to analyze RT-qPCR data, which was normalized using *Gapdh* or *β-actin*. The expression of specific genes in germ cells and granulosa cells was normalized using *Ddx4* and *Amhr2*, respectively. The primers information was listed in Supplementary Table 3.

**Western blot analysis**
Ovarian mixed samples of each group were collected, and treated with the RIPA solution (Beyotime, P0013C, Shanghai, China) to obtain protein lysates. The SDS-PAGE isolated proteins from protein lysates and transferred them to PVDF membranes (Millipore, IPVH00010, Germany) for subsequent steps: incubated with primary antibody for 12-h at 4 °C after blocked; secondary antibody for 60 min at 25 °C after washed. Antibody information are listed in Supplementary Table 2. Following that, chemiluminescence were performed using a BeyoECL Plus Kit (Beyotime, A0018), as directed by the manufacturer. The expression level of target protein was represented by calculating the ratio of gray value of target protein to the corresponding housekeeping protein (GAPDH, β-ACTIN, H3, H4, or DDX4) by using the AlphaView SA software (ProteinSimple, San Jose, CA, USA).

**Spindle staining**
To investigate whether the defects in the primordial follicle formation found in maternal HFD mode during pregnancy affect oocyte spindle formation, we examined the abnormal spindle rate of oocytes in three-week-old offspring. Cumulus-oocyte complexes (COCs) and MII stage oocyte were obtained from the ampulla of mice after intraperitoneal injection of PMSG for 48-h and HCG for half a day. After fixation for 0.5-h at 25 °C, the spindle was stained as follows: 0.5% Triton X-100 penetrates for 0.4-h; blocked with blocking buffer for 1-h at 25 °C; incubated with anti-α-tubulin antibody (Supplementary Table 2) for 3-h; incubated with FITC-labeled secondary antibodies (Supplementary Table 2) for 1-h at 25 °C after washed. Chromosomes were stained with Hoechst 33342 (Beyotime, C1022) and images were examined with a Nikon fluorescence microscope (A1, Japan).

**In vitro fertilization (IVF)**
In addition, we examined the embryonic development rate of oocytes in three-week-old offspring. COCs were collected from each group as described in the step of spindle staining. Subsequently, after HTF medium (Aiber Biotechnology Co., Ltd, M1130, Nanjing, Chian) balancing sperm in the caudal epididymis of male mice, the COCs of each group was mixed in TYH medium (Aiber Biotechnology Co., Ltd, M2030) with capacitated sperm for the fertilization procedure in vitro. Successful fertilization was characterized by the appearance of two pronuclei. Oocytes were transplanted into KSOM medium (Aiber Biotechnology Co., Ltd, M1430) and development rates were analyzed over the following days to the indicated stages.

**Transmission electron microscopy analysis**
In this study, three ovaries from each group were used and fixed in 2.5% glutaraldehyde for 12-h at 4 °C as biological replicates, followed by embedding, sectioning and staining procedures using standard transmission electron microscopy methods[63]. Finally, the morphology of mitochondria in ovarian cells was observed and analyzed by HT7700 transmission electron microscopy (Hitachi, Japan).

**Metabolomics sequencing and analysis**
Serum samples were separated from the blood of maternal mice in mND and mHFD group with 9 replicates per group. UHPLC-Q/TOF-MS in BMKCloud (www.biocloud.net) was used to profile the metabolomics of the supernatants. In brief, using a 1290 LC infinity set-up (Agilent technologies, USA), liquid phase chromatographic fractionation were performed on processed serum samples. Metabolites were determination by gradient elution using the mobile phase of positive and negative ion modes as described in Zeng's articles[35].

MassHunter Workstation Software (Agilent Technologies) translated raw data files to the mzdata format, which was then processed by the XCMS suite (version 3.2). The orthogonal partial least squares-discrimination analysis (OPLS-DA) were performed to determine the discrimination of variables. Metabolites that met the following criterion of nominal *P*-values less than 0.05, variable importance in the projection (VIP) greater than 1 and fold change greater than 2 were considered significant differences metabolite. The altered metabolic pathways in connection to the identified SDMs were determined using MetaboAnalyst 5.0[64].

**Measurement of thiamine (Vitamin B1, VB1) content**
Maternal serum, intestinal fecal and offspring ovary levels of VB1 content was determined using General VB1 ELISA kit (Signalway Antibody Co. Ltd, EK4294, Maryland, USA). Firstly, the serum is diluted with the sample diluent at a ratio of 1:3. Each 0.1 g fecal samples and 1 offspring ovary were diluted with 100 μL and 10 μL sample diluent, respectively, and the supernatant was collected. The subsequent reaction steps were performed using 50 μL/well of sample extract and

standard strictly as directed by the manufacturer. The optical density at 450 nm was calculated using an ELISA instrument (450 nm), and the thiamine content was determined using the method derived from the standard curve's linear regression.

### VB1 deficiency and supplementation models

To investigate the role of maternal VB1 levels in fetal phenotypes, we designed two models of maternal VB1 deficiency (Fig. 2e): mTA model, 60 mg/kg VB1 antagonist Amprolium (APL, Selleck, S4144, Shanghai, China) was given intragastric (i.g.) administration starting from E7.5 on the basis of normal diet feeding during pregnancy; mTD model, thiamine deficiency diet (Dyets, D201218, Wuxi, China) feeding starting from E14.5 during pregnancy. The timing of mTA and mTD feeding was determined based on preliminary trials of VB1 concentration (Supplementary Fig. 4a). In addition, we designed the model for maternal VB1 supplementation (mHFD+VB1): 0.25 mg/kg thiamine (Selleck, S3211) was intraperitoneally (i.p.) injected starting from E14.5 on the basis of high-fat diet feeding during pregnancy (Fig. 2e). On postnatal day 3, ovarian from female offspring and maternal serum samples were collected for subsequent experimental analyses.

### 16 S rDNA amplicon sequencing and analysis

Fecal samples were collected from the respective groups before the sequencing and stored at −80°C. The construction and sequencing of a 16 S rDNA library were performed using the Biomarker Technologies platform (www.biomarker.com.cn) is as reported in our recent publications[65]. Finally, R software was used to visualize the principal coordinate analysis (PCoA), taxonomy, and heatmap of differential abundance.

### Antibiotic treatment and fecal transplantation experiments

Six weeks recipient female C57BL/6 mice fed were treated with antibiotics including 0.5 g/L vancomycin, 1 g/L neomycin sulfate, 1 g/L metronidazole, and 1 g/L ampicillin for one week, and then for fecal transplant. Fresh fecal samples of pregnant donor mouse were collected from mND and mHFD groups, and prepared with saline at 0.4 g/mL. The recipient mice treated with antibiotics were allocated into two groups (mND-FMT and mHFD-FMT) at random, and they were given 100 μL 0.4 g/mL fecal samples from mND mice or mHFD mice every other day. One week after inoculation, the recipient mice were impregnated at 5 p.m. and fecal transplant were continued after vaginal plug appeared the second day. The fecal microbiota of recipient mice was transplanted for four weeks for subsequent experiments.

### Hematoxylin-eosin (H&E) staining

The maternal intestinal tissues (jejunum and lleum), and the ovaries of offspring at 7 months of age were fixed in bouins's solution (sigma, HT10132). Perform paraffin embedded, cut into 5 μm sections as described above. After deparaffinization, subsequently stained with H&E for histopathological analysis.

### Ovary culture in vitro, siRNA and inhibitor treatment

For ovary culture in vitro: After isolation ovaries from newborn mice, the ovaries were cultured on 3D-culture membrane (Millipore, PIHA01250) for 72-h. The culture medium was DMEM/F12 (Gibco, C11330500BT, Beijing, China) supplemented with 10% FBS (Gibco, 10099-141), 1% double-resistance (HyClone, SV30010, Beijing, China). Ovarian tissue was incubated in standard cell incubator, which were replaced daily with fresh media.

For siRNA treatment: Before transfection, 1.5 μL TransIntro™ Transfection Reagent (TransGen Biotech, FT301, A) and 2 μL 20 μmol/L *Gapdh*, *Pdha1*−213/678/1051, *Slc19a2*−623/954/1547, *Slc19a3*−236/355/1059, or scrambled-siRNA (Scr.siRNA, GenePharma, Shanghai, China, B) were combined with 50 μL basal medium,

respectively, and at 25 °C for 5 min. The sequence information was listed in Supplementary Table S4. Subsequently, liquid A was added to liquid B and gently mixed, and let stand for a quarter of 1-h at 25 °C before adding to the medium. After 7 hours, we changed the medium to fresh medium. The relative mRNA and protein levels were measured 48 h and 72 h later, respectively.

For inhibitor treatment: Stock solutions of 200 mM PDH inhibitor CPI-613 (Selleck, S2776) were prepared in DMSO, and then save in light-proof −20 °C. Work solutions of 100 μM CPI-613[66] were prepared in fresh medium. Stock solutions of 100 mM VB1 antagonist APL were prepared in PBS, and then save in −20 °C. Work solutions of 500 μM and 1 mM APL were prepared in fresh medium.

### PDH activity assay

The PDH activity Assay Kit (Solarbio, BC0385, Beijing, China) was used to assess PDH activity. PDH can not only catalyze dehydrogenation of pyruvate, but also reduce 2.6-DCPIP, resulting in reduced light absorption at 605 nm to achieve the detection effect. The optical density after 0 (A1) and 1 (A2) min was determined using ELISA instrument to 605 nm, respectively. The PDH activity assays were all performed using delta A (A1 - A2) in accordance with the manufacturer's instructions.

### Measurement of Acetyl-CoA level

The Acetyl-CoA Assay Kit (Sigma, MAK039) was used to assess the acetyl-CoA concentration of the ovary. Operation steps and evaluation methods are carried out according to the manual. In brief, ovary samples were immediately frozen (liquid N2) and crushed before being deproteinized by perchloric acid (Sigma, 34288) precipitation. After removal of insoluble particles by centrifugation (8000 x g), the reaction step was performed follow manufacturer's instructions. The 6 points standard curve were mapped using 0, 0.2, 0.4, 0.6, 0.8 and 1 nM acetyl-CoA. The optical density was determined using ELISA instrument (λex = 535/λem = 587 nm), and the Acetyl-CoA level was determined using the formula obtained from the standard curve's linear regression.

### Measurement of ATP level

The Enhanced ATP Assay Kit (Beyotime, S0027) was used to determine ATP levels. Briefly, sample fluid is obtained by processed the ovarian tissue by the manufacturer. In each assay, the five-point standard curve (0, 0.1, 1, 5 and 10 μL ATP) was created, and the ATP content was determined using the formula obtained from the standard curve's linear regression.

### Reactive oxygen species (ROS) evaluation

As mentioned in our prior work[62], intracellular ROS analysis was carried out using the ROS Assay Kit (Beyotime, S0033S). Briefly, freshly ovaries from each group were collected and treated for 0.5-h at 37 °C in the dark with ROS dyeing reagent. After that, the tissues were rinsed twice with PBS and the fluorescent intensity of ovaries from each group was instantly quantified using a Nikon fluorescence microscope.

### Ovarian cell nuclear isolation

Isolation of nuclei was performed using the commercially available nuclei extraction kit (Solarbio, SN0020). Briefly, ovarian were collected and extracted in the presence of lysis buffer after adding Reagent A. The nuclei were recovered after centrifuging the suspension at 700 × g for 5 min, and 500 μL Lysis Buffer was used to resuspend the nuclei. The nuclei were then resuspended with 0.5 mL lysis buffer and transferred to 0.5 mL ice precooled medium buffer. Finally, purer nuclei at the bottom of the tube were obtained by centrifugation (1000 x g). Western blotting was used to detect the nuclear purity and the nuclei presence of PDC-E1 (Supplementary Table 2).

## EdU proliferation assay

The BeyoClick™ EdU Cell Proliferation Kit (Beyotime, C0071S) was used to determine the positive-cells number of granulosa cells in ovary. After EdU treatment, the serial sections of each ovary were labeled germ cells with anti-FOXL2 antibodies (Supplementary Table 2) following the above IF treatment procedure. Then, the sections were stained of EdU following the manufacturer's instructions for double-labeling. Nuclei were visualized with DAPI. ImageJ software (NIH, Bethesda, MD, USA) was used to score the FOXL2-EdU double-positive cells in each ovary.

## Assay for transposase-accessible chromatin (ATAC) sequencing

Cell nuclei preparation: follow the steps for single cell sequencing sample preparation described above, the ovaries were collected from the mND and mHFD offspring mice at P3. Each group had two replicates, and 50 ovaries were collected from each replicate.

The ovarian tissue was digested into a single cell suspension by collagenase, and then differentially adherent culture was performed after inoculation. Finally, the non-adherent germ cells and ovarian tissue were removed to obtain granulosa cell suspension that met sequencing criteria for cell survival rate. The three times wash with PBS and cell pellets were lysised for ten min at four degrees on the rotating mixer. Tn5 transposed DNA was purified using AMPure DNA magnetic beads. The library was performed with an amplification program running on Agilent Tapestation 2200 after the appropriate number of PCR cycles was established.

Trimmed reads were aligned to refence genomes using Bowtie2 (version 2.2.6)[67], and removeChrom was used to delete reads that mapped to the mitochondrial genome of the genome. PCR duplicates were removed and the number of mapped reads down-sampled to a standard of ~20 M using Picard (version 1.126) for the samples. Call peaks on replicates, self-pseudoreplicates using MACS2[68]. We run IDR assessment on peaks between each sample and self-pseudoreplicates with IDR threshold set to 0.05. The R package DESeq2 (version 1.28.1) was used to obtain a differentially accessible regions (DAR) with |log2 fold change| less than 0.5 and nominal $P$-value more than 0.05 from the consensus peak set. The R package "ClusterProfiler" was performed GO enrichment with default settings. Using the IGV (http://software.broadinstitute.org/software/igv/), the ATAC-sequencing tracks were examined.

## Cell culture

Primary mouse granulosa cells were extracted from the ovaries tissue of mice at postnatal day 12 (P12) in accordance with the research of Zhang et al.[69]. Briefly, the ovarian tissue was digested into a single cell suspension and non-adherent germ cells and ovarian tissue were removed by differential adherent culture after inoculation 12-h. Primary mouse granulosa cells were maintained on DMEM medium (HyClone, SH30022.01) supplemented with 10% FBS and 1% double-resistance in standard cell incubator.

## Cell cycle analysis

The cell cycle detection kit (Beyotime, C1052) was used to score the cell cycle distribution. The collected cells were fixed with 70% ethanol at 4 °C overnight, and then stained with 0.05 mg/mL PI and 0.1 mg/mL RNase-A for 15 min at 37 °C, according to the instructions. Finally, the FACS Calibur flow cytometer (Cytoflex LX, China) was utilized to measure least 10,000 cells per sample and analyzed using FlowJo V10 software.

## Statistical analysis

The data (mean ± SD) was analyzed using the GraphPad Prism 8 analysis software. The two-tailed unpaired Student's $t$ test was performed to compare the two sets of data. One-way analysis of variance (ANOVA) was performed to evaluate datasets with three or more groups, followed by Tukey's test for multiple comparisons. $P$-value less than 0.05 was statistically significant difference and $P$-value less than 0.01 was highly significant difference.

## Reporting summary

Further information on research design is available in the Nature Portfolio Reporting Summary linked to this article.

## Data availability

All relevant data are available within the article and its supplementary information/Source Data. The data of offspring ovarian single cell RNA sequencing, maternal gut microbiome, and offspring ovarian granulosa cells ATAC-sequencing that support the findings of this study has been deposited in the Genome Sequence Archive (GSA, https://ngdc.cncb.ac.cn/gsa) under the accession number CRA005316, CRA005301, and CRA005298, respectively. Moreover, the data of maternal serum metabolomics have been deposited in the Open Archive for Miscellaneous Data (OMIX, https://ngdc.cncb.ac.cn/omix) under the accession number OMIX721 (https://ngdc.cncb.ac.cn/omix/release/OMIX721). Furthermore, in this study, the analysis was supplemented with the utilization of the KEGG (https://www.genome.jp/kegg) and GO (https://geneontology.org) databases for comprehensive functional enrichment analysis of the data. Source data are provided with this paper.

## Code availability

The R code using for data processing is publicly available at GitHub at https://github.com/ZZlab412/Murine-thiamine-DATA/tree/V1. The DOI is https://doi.org/10.5281/zenodo.8435158[70].

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

## Acknowledgements

This study was supported by the Young Talents of Science and Technology in Universities of Inner Mongolia Autonomous Region (NJYT23013 to Y.Zhou), the Central Government Guides Local Science and Technology Development Fund Projects (2022ZY0188 to ZT), the National Natural Science Foundation of China (32260180 to T.Z.; 32260181 to Y.Zhou; 32000582 to T.Z.; 31900595 to Y.Zhou), the Science and Technology Major Project of Inner Mongolia Autonomous Region of China to the State Key Laboratory of Reproductive Regulation and Breeding of Grassland Livestock (2019ZD031 to Y.Zhou; 2021ZD0048 to T.Z. and Y.Zhou).

## Author contributions

T.Z. and Y.Zhou conceived the project; W.X.L., H.N.L., Z.P.W., Q.G., Y.Zhang and Y.F.L. performed the experiments and analyzed the data; W.X.L. and Q.G. contributed to the bioinformatics analysis; Y.Zhou and T.Z. wrote the manuscript; Y.Zhou, T.Z. and W.S. contributed to the manuscript revision and discussion. All the authors read the manuscript and approved the final manuscript.

## Competing interests

The authors declare no competing interests.
