## [Peer Review File · Nature Communications]

REVIEWER COMMENTS

Reviewer #1 (Remarks to the Author):

The manuscript submitted to Nature Communications entitled “Maternal vitamin B1 is a determinant for the fate of primordial follicle formation in offspring” investigates the effects maternal high fat diet on primordial follicle formation in female offspring and links effects to vitamin B1 levels. The authors find that a maternal high fat diet (mHFD) reduces follicle formation in female neonates. They use scRNAseq to identify differentially regulated genes in the ovaries of female neonates, find among other things, one class of genes linked to mitochondrial function and go on to show that mHFD causes oocyte mitochondrial dysfunction in offspring. They also analyze the metabolomic profile of the maternal serum from mice on a normal diet and HFD and find differences in vitamin processing including vitamin B1. They go on to show that maternal vitamin B1 deficiency reduces follicle formation. They next turn to the maternal gut microbiome and find that mHFD alters the microbiome and alters vitamin B absorption. They also address acetyl coA production in oocytes and granulosa cell proliferation. They finish with analyzing samples from human patients with gestational diabetes and find they have lower vitamin B1 levels. This work in mice addressing effects of maternal nutritional status on the ovarian reserve of female offspring is important for understanding effects maternal nutrition on human female fertility. My major concern is that there is a huge amount of data presented in each figure and there several stories that can be extracted from the data. The way the information is presented makes it hard to follow. Most of the figures could be split into two figures and it might be beneficial to split this manuscript into at least two separate papers. I expand upon major issues below. Additionally, several minor issues also need to be addressed.

Major issues:

1. I feel that there are too many large data sets for one paper: scRNAseq, metabolomic profiling, ATAC seq, and microbiome analysis.
2. For the first section of the results and figure 1b-g it is unclear which mouse strain is being used. It is not stated but from the statement in line 105 “in addition, we performed the experiments using ICR mice” it is assumed that the experiments above were in B6 mice.
3. Line 108-109, 21 day old and 7 month old mice had fewer primordial follicles but the authors do not address the cause of this loss. Do more oocytes die or do more follicles get activated? This needs to be addressed.
4. Line 110 and 111, the authors need to be more specific than just saying loss of oocyte quality and developmental potential.
5. Line 125-127, need to be more specific about how the cell types were identified.
6. Line 129-130, need to elaborate/explain “ branched point from three states...”

7. Line 164-166, only three of these 6 are related to water soluble vitamin. Retinyl ester is vitamin A (fat soluble), dodecanoic acid is a fatty acid, and calcitrol is vitamin D (fat soluble).
8. Line 168-169, age of offspring and mother fig 3e and 3g ---says p3 for 3g but no mention in figure legend or elsewhere.
9. Figure 3 g-i, what age is shown (P3 for figure 3f) and was the ND and mHFD repeated or are these data from the original experiment in Figure 1.
10. Line 179-180, how was fetal development enhanced?
11. Line 186-187, explain why you are using the RNA sequencing.
12. Line 199-201, explain the rationale behind the fecal microbiota transplantation experiment.
13. Figure 5h,i, image very small and not convincing.
14. Figure 6m, n, how were granulosa cells identified also images are very small and not convincing.
15. How were participants identified for follow up and were they from an ongoing study?
16. Line 320, define steatosis and how to identify in Figure 7d
17. Line 321, the levels of GLU and TG do not look similar in ND and mHFD.
18. Figure 1a, what are the 2 green rectangles?
19. Line 1057, what is in red, why not use a granulosa cell marker?

Minor issues:

1. The whole document needs to be checked for correct spelling and grammar, some instances noted here.
2. Line 99, "examined by using immunostained" should be "examined by immunostaining"
3. Line 128, the R package, Monocle (instead of just 'monocle).
4. Line 134-135, I think the wording is not correct---you didnt enrich the genes?
5. Line 135-137, again the wording is not correct-"we gained insight into the GO terms"?
6. Line 172, "attempted to design" either you designed it or you didn't.
7. Line 177, 178, delete "the development"
8. Line 205-207 does not make sense.
9. Line 270, do they really mean "lost"?
10. The references are separated, some are on page 22 and some on page 39.

11. Line 578, what is after breakfast fasting?
12. Line 825 and 830 what are PG cells?
13. Figure 2a, bracoding should be barcoding.
14. Figure 7j,k, l, relative is spelled wrong.

Reviewer #2 (Remarks to the Author):

This is a very well written and comprehensive study, including noteworthy results. The research question is novel and the potential implications of the results are impactful. The study includes a multi-faceted methodological approach, which is elegant and well designed. Overall, results support the conclusions made throughout the manuscript. Yet, there are a few points that require clarification:

1. The authors results show that maternal high-fat intake during pregnancy induces VB1 insufficiency which, in turn, causes a low number of primordial follicles in the offspring. Did the authors test whether ovarian function in the F1 or F2 generation? If data are not available, please discuss.
2. All figures are beautifully presented; yet, in many cases the authors are comparing multiple groups. Do they use multiple comparisons? The p values are highly significant, which makes me wonder whether adjustment for multiple comparison were used.
3. The fecal microbiota transplanted experiments require further description. Can you please elaborate in the study groups and controls? Were pregnant mice (without HFD) transplanted with fetal microbiota of dams with HFD and vice versa? This is not clear in Figure 4H. Also, what are the controls?
4. Maternal age was significantly different between GMD and HC women. Did the authors adjusted for maternal age?
5. In general, the manuscript is beautifully written and the results are presented in a very compelling manner.

Reviewer #3 (Remarks to the Author):

This study intends to uncover the role of vitamin B1 deficiency in dam gut attributing to growth failure of primordial follicle in maternal HFD-induced fetuses, the major findings include: 1) identified pregnant HFD induces dam and fetal serum vitamin B1 deficiency, which is associated with dam microbiome disruption; 2) vitamin B1 deficiency inhibits ovary PDH activity, the decrease of acetyl-CoA further impairs histone acetylation of proliferative genes in granulosa cells; 3) gestational diabetes also decrease serum vitamin B1 in dams.

The manuscript is readable and the study is interesting, but there are several major concerns required to address:

1) The pregnant HFD impacts on fetal ovary development should be due to multidimensional factors, such as excessive of saturated and unsaturated fatty acids intake, thus it is vital to block vitamin B1 receptor in fetal follicular to determine the actual impacts of vitamin B1 deficiency on primordial follicle maturation and gene acetylation.

2) It is not convincing how gestational diabetes are able to link to pathology of pregnancy HFD feeding.

3)The author examined 16S ribosomal DNA (rDNA) amplicon

sequencing, and find that the 43 bacteria were altered at the genus level in the mHFD group compared with the mND group. Further, the author mentioned that these findings above prompted us to consider that these microbiotas appeared to contribute to VB1 metabolism and absorption. However, the author did not use metagenomics methods to check whether or not these microbiome can synthesize vitamin,therefore, the author should explore the mechnism underlying the VB1 systhesis by gut microbe.

4)The fecal microbiota transplantation (FMT) increased the levels of VB1 and the protein expression levels of SLC19A3 was decreased in intestinal digesta from mHFD recipient mice (mHFD-FMT) compared with mND recipient mice (mND-FMT), respectively (Fig.4i, j).I just wonder that which microorganisms plays a key role in the synthesis of VB1, so, more experiments are needed to test in this section.

Reviewer #4 (Remarks to the Author):

This manuscript describes the potential role for reduced thiamine levels as a mediator of alterations in oocyte development in female offspring of female mice fed a HFD upon initiation of pregnancy. The authors demonstrate using microbiome analysis and fecal transplantation that maternal HFD induces a change in the microbiome, associated with reduced expression of the thiamine transporter in maternal intestine and parallel changes in thiamine levels in the maternal circulation. These changes are associated with altered developmental trajectories of the developing fetal germ cells and expression of nuclear-encoded mitochondrial genes and mitochondrial morphology. Modulation of thiamine levels and/or signaling reversed these effects, which were also associated with parallel changes in post-translational histone acetylation and chromatin accessibility. Finally, the authors present data that thiamine levels are reduced in humans with gestational DM.

In general, the manuscript is interesting and the reversal experiments are quite striking. However, the manuscript is challenging to read as results are not clearly described. Limitations of the study are not well-delineated. Moreover, the human data are not well-linked to the rest of the paper. More information is needed about the women with GDM. Were they obese as compared with the controls? Matched for age? Why not just study a cohort of women with obesity since that is what the rest of the manuscript is focused.

Were levels of butyrate or acetate changed in the maternal circulation in the mice?

Please provide information about weight gain during pregnancy, litter size, birth weight in the mice, including chow, HFD, thiamine deficiency, vitamin B1 antagonist compound treatment, and B1 supplementation groups.

For each of the omics analysis please provide information about whether data were corrected for multiple comparisons, and which method was used to do so?

Figure 3D – Please color by group. It appears that there is not a spectrum of B1 levels but rather stratification by group, in which case the correlation has a someone different conclusion.

Minor:

Line 662 – what is “gray value of target proteins”?

Line 840 – what does “conforming to the standard of cell survival rate” mean?

Line 879 – $p < 0.01$ should not be considered “extremely significant difference.”

**REVIEWER COMMENTS**

**Reviewer #1** (Remarks to the Author):

**The manuscript submitted to Nature Communications entitled “Maternal**
**vitamin B1 is a determinant for the fate of primordial follicle formation in**
**offspring” investigates the effects maternal high fat diet on primordial follicle**
**formation in female offspring and links effects to vitamin B1 levels. The authors**
**find that a maternal high fat diet (mHFD) reduces follicle formation in female**
**neonates. They use scRNA-seq to identify differentially regulated genes in the**
**ovaries of female neonates, find among other things, one class of genes linked to**
**mitochondrial function and go on to show that mHFD causes oocyte**
**mitochondrial dysfunction in offspring. They also analyze the metabolomic**
**profile of the maternal serum from mice on a normal diet and HFD and find**
**differences in vitamin processing including vitamin B1. They go on to show that**
**maternal vitamin B1 deficiency reduces follicle formation. They next turn to the**
**maternal gut microbiome and find that mHFD alters the microbiome and alters**
**vitamin B absorption. They also address acetyl-coA production in oocytes and**

**granulosa cell proliferation. They finish with analyzing samples from human**
**patients with gestational diabetes and find they have lower vitamin B1 levels.**
**This work in mice addressing effects of maternal nutritional status on the**
**ovarian reserve of female offspring is important for understanding effects**
**maternal nutrition on human female fertility. My major concern is that there is a**
**huge amount of data presented in each figure and there several stories that can**
**be extracted from the data. The way the information is presented makes it hard**
**to follow. Most of the figures could be split into two figures and it might be**
**beneficial to split this manuscript into at least two separate papers. I expand**
**upon major issues below. Additionally, several minor issues also need to be**
**addressed.**

**Response:** We appreciate the reviewer's concern. We have revised the manuscript to
ensure that the data is presented in a clear and concise manner, with appropriate
emphasis placed on the most important findings. Additionally, we will carefully
consider the issues raised and make revisions to ensure the manuscript meets the high
standards expected for publication in this journal.

**Major issues:**

**Question 1: I feel that there are too many large data sets for one paper: scRNA-**
**seq, metabolomic profiling, ATAC-seq, and microbiome analysis.**

**Response:** Thank you for your concern. Our study employed a comprehensive multi-
omics approach to investigate the interplay between maternal vitamin B1 (VB1)
levels, gut microbiota, and molecular mechanisms involved in primordial follicle
formation in offspring. We believe that each technique is necessary to reveal the
intricate molecular dynamics of the effects of maternal VB1 on primordial follicle
formation in offspring.

**Question 2: For the first section of the results and figure 1b-g it is unclear which**
**mouse strain is being used. It is not stated but from the statement in line 105 “in**
**addition, we performed the experiments using ICR mice” it is assumed that the**

**experiments above were in B6 mice.**

**Response:** We appreciate the reviewer's attention to detail. We have made the
revisions to ensure that it is clear which mouse strain was used for each set of
experiments. We have added a statement in the methods section of lines 457: "*With*
*the exception of ICR mice utilized in the Supplementary Fig.1e-h experiment depicted,*
*C57BL/6 mice were employed in all other mouse experiments conducted within this*
*study.*".

**Question 3: Line 108-109, 21 day old and 7 month old mice had fewer primordial**
**follicles but the authors do not address the cause of this loss. Do more oocytes die**
**or do more follicles get activated? This needs to be addressed.**

**Response:** Thanks for your professional review work on our manuscript. At 3 weeks
and 7 months, we detected a significant decrease in the number of total follicles,
primordial follicles, primary follicles, and secondary follicles in offspring ovaries
from mHFD group in comparison to mND group (Figure a, b). Importantly, we
observed that the mHFD group had a lower number of growing follicles in their
offspring ovaries at these time points (Figure c). The decrease in primordial follicle
numbers is not attributed to excessive activation, as indicated by our findings. In
addition, primordial follicle assembly largely determines the ovarian reserve that will
be available to support the reproductive life span (J. J. Wang et al., 2020), and any
aberrations during this process can lead to premature depletion due to follicle death
(Ge, Li, Dyce, De Felici, & Shen, 2019; Kato et al., 2019). Together, our findings
indicate that maternal HFD adversely affects the primordial follicles formation in
offspring, resulting in a decline in ovarian reserve.

In addition, we add a description of this result at line 104: "*Moreover, there were*
*fewer primordial follicles and growing follicles in the ovaries of 3-week-old and 7-*
*month-old offspring from mHFD mice (Fig.1h and Supplementary Fig.1d), indicating*
*that the defects of ovarian reserves had a long-range effect.*".

**Reference:**

Ge, W., Li, L., Dyce, P. W., De Felici, M., & Shen, W. (2019). Establishment and depletion of the

ovarian reserve: physiology and impact of environmental chemicals. *Cell Mol Life Sci*, 76(9),
 1729-1746. doi:10.1007/s00018-019-03028-1
 Kato, Y., Iwamori, T., Ninomiya, Y., Kohda, T., Miyashita, J., Sato, M., & Saga, Y. (2019). ELAVL2-
 directed RNA regulatory network drives the formation of quiescent primordial follicles.
 *EMBO Rep*, 20(12), e48251. doi:10.15252/embr.201948251
 Kobayashi, N., Orisaka, M., Cao, M., Kotsuji, F., Leader, A., Sakuragi, N., & Tsang, B. K. (2009).
 Growth differentiation factor-9 mediates follicle-stimulating hormone-thyroid hormone
 interaction in the regulation of rat preantral follicular development. *Endocrinology*, 150(12),
 5566-5574. doi:10.1210/en.2009-0262
 Wang, J. J., Ge, W., Zhai, Q. Y., Liu, J. C., Sun, X. W., Liu, W. X., . . . Shen, W. (2020). Single-cell
 transcriptome landscape of ovarian cells during primordial follicle assembly in mice. *PLoS*
 *Biol*, 18(12), e3001025. doi:10.1371/journal.pbio.3001025

 **Figure Legends:** **a** Quantification of the number of oocytes and different classes of follicles in
 3 weeks ovaries of offspring from mND and mHFD groups ($n = 13$ biologically independent
 repeats). **b** Quantification of the number of oocytes and different classes of follicles in 7
 106 months ovaries of offspring from mND and mHFD groups ($n = 8$ biologically independent
 repeats). **c** Number of growing follicles at 3 weeks and 7 months in offspring ovaries from
 mND and mHFD groups ($n \geq 8$ biologically independent repeats). Student's *t* test (two-tailed)
 was used for statistical analysis; n.s., not significant.

 **Question 4:** Line 110 and 111, the authors need to be more specific than just
 saying loss of oocyte quality and developmental potential.

**Response:** Thank you for your suggestion. We have revised the manuscript to provide

more specific details regarding the loss of oocyte quality and developmental potential
in line 113: “Specifically, the rate of aberrant spindles and misaligned chromosomes
was significantly increased in the mHFD group (Supplementary Fig.2a, b). In
addition, the abnormal proportion of 2-cell embryos, 4-cell embryos, and blastocysts
was significantly increased after in vitro fertilization in the mHFD group

(Supplementary Fig.2c, d).”.

**Question 5: Line 125-127, need to be more specific about how the cell types were**
**identified.**

**Response:** Thanks for your professional review work on our manuscript. In this study,
ovarian cells were divided into 12 cell clusters based on the similarity of their gene
expression profiles by means of unsupervised clustering algorithms (such as k-means
or hierarchical clustering) in scRNA-seq. These resulting clusters are then annotated
based on known marker analysis to assign putative cell type identities, such as *Ddx4*-
and *Dazl*- positive cell cluster defined as germ cells.

Based on your comments, we have revised the manuscript to provide a more
detailed explanation of how cell types are identified in line 184, “*Corresponding to*
*known clustering of different cell types from the neonatal ovary (J.-J. Wang et al.,*
*2019), we identified six major cell types including germ cells (clusters 4, 6) with*
*Ddx4 and Dazl expression, granulosa cells (clusters 0, 2, 3, 10) with Amhr2 and Kitl*
*expression, stromal cells (clusters 1, 9, 11) with Mfap4 and Colla1 expression,*
*endothelial cells (cluster 8) with Egf17 and Aplnr expression, erythrocytes (cluster 5)*
*with Alas2 and Rhd expression, and immune cells (cluster 7) with Cd52 and Cd53*
*expression (Fig.3b and Supplementary Fig.7d).”.*

**Reference:**

Wang, J.-J., Ge, W., Zhai, Q.-Y., Liu, J.-C., Sun, X.-W., Liu, W.-X., . . . De Felici, M. J. b. (2019).
Transcriptome Landscape Reveals Underlying Mechanisms of Ovarian Cell Fate
Differentiation and Primordial Follicle Assembly. 803767.

**Question 6: Line 129-130, need to elaborate/explain “branched point from three**
**states...”.**

**Response:** We appreciate your professional comments. In the study, the "branched
point from three states" refers to the identification of a point in the pseudotime
analysis of germ cells where three different states diverge, each characterized by a
unique gene expression pattern.

R Package “Monocle” is a software tool that can perform pseudotime analysis of

single-cell RNA sequencing data (Trapnell et al., 2014). Specifically, “Monocle”
assigns a pseudotime value to each germ cells based on its gene expression profile.
Then, “Monocle” constructs a minimum spanning tree to connect cells with similar
gene expression profiles and orders them based on the pseudotime value. This
ordering allows “Monocle” to identify the branch points where germ cells diverge into
distinct states. Based on its gene expression profile, “Monocle” can compare the
pseudotime trajectories of different groups (e.g., mND vs. mHFD) of germ cells to
identify differences in gene expression patterns that reflect changes in cellular state.
Further analysis of these states could provide potential insight into the mechanisms
underlying the differences between the mND and mHFD groups.

**Reference:**

Trapnell, C., Cacchiarelli, D., Grimsby, J., Pokharel, P., Li, S., Morse, M., . . . Rinn, J. L. (2014). The
dynamics and regulators of cell fate decisions are revealed by pseudotemporal ordering of
single cells. *Nature biotechnology*, 32(4), 381.

**Question 7: Line 164-166, only three of these 6 are related to water soluble**
**vitamin. Retinyl ester is vitamin A (fat soluble), dodecanoic acid is a fatty acid,**
**and calcitrol is vitamin D (fat soluble).**

**Response:** Thank you for your suggestion. Our intention was to highlight the six
metabolites that were significantly altered by HFD enrichment in the pathway of
vitamin digestion and absorption. We have revised in our manuscript to accurately
describe it in line 132: “*In addition, the pathway of vitamin digestion and absorption*
*was remarkably prominent (Fig.2b), and six metabolites within this pathway showed*
*dramatic changes in the mHFD group (Fig.2c).*”.

**Question 8: Line 168-169, age of offspring and mother fig 3e and 3g ---says p3 for**
**3g but no mention in figure legend or elsewhere.**

**Response:** Thank you for your suggestion. In Fig.2d and 2f, the data represents
maternal serum samples and offspring ovarian samples collected from each group at
postnatal day 3 (P3). The time information collected by this sample should be
included in the legend of the manuscript.

To address this issue, we modified the figure legend to clearly state that the data
in Fig.2d and 2f corresponds to samples collected at postnatal day 3. We also mention
in the Materials and Methods section of the manuscript when samples were collected
in relation to these figures.

**Question 9: Figure 3 g-i, what age is shown (P3 for figure 3f) and was the ND and**
**mHFD repeated or are these data from the original experiment in Figure 1.**

**Response:** We appreciate your professional comments. In Figure 3 g-i (corresponding
to Fig.2f-h in the revised version), the age shown is postnatal day 3 (P3), as indicated
in Fig.2e. We modified the figure legend and manuscript text to clearly state that the
data in Fig.2f-h correspond to samples collected at P3. It is important to note that the
data presented in Fig.2 (mND and mHFD conditions) are from independent
experimental replicates and not a replication of the original experiment displayed in
Fig.1.

**Question 10: Line 179-180, how was fetal development enhanced?**

**Response:** Thanks for your comment. We have revised in our manuscript to
accurately describe it in line 140: *“After exposure to 60 mg/kg of the VBI antagonist,*
*Amprolium (mTA), for 14 days or a VBI deficiency-diet (mTD) fed for 7 days during*
*pregnancy, the levels of VBI in maternal serum, the development of primordial follicle*
*formation, and the growth indexes of fetus closely resembled those observed in the*
*pregnant maternal HFD model (Fig.2f-h and Supplementary Fig.4b-d). Furthermore,*
*VBI supplementation during pregnancy promotes maternal serum VBI levels and*
*primordial follicles formation as well as the growth indexes of fetus (Fig.2f-h and*
*Supplementary Fig.4b-d). Collectively, we indicated that that maternal VBI during*
*pregnancy protected offspring from impaired primordial follicle formation.”.*

**Question 11: Line 186-187, explain why you are using the RNA sequencing.**

**Response:** Thanks for your comment. We would like to clarify that in Line 186-187
(corresponding to line 152 in the revised version), we performed 16S rDNA

sequencing to characterize the composition and diversity of the gut microbiota instead
of RNA sequencing. This analysis aimed to explore the potential relationship between
gut microbiota dysbiosis and VB1 absorption.

**Question 12: Line 199-201, explain the rationale behind the fecal microbiota**
**transplantation experiment.**

**Response:** Thanks for your professional review work on our manuscript. The
rationale behind the fecal microbiota transplantation (FMT) experiment is based on
the concept that the gut microbiota plays a crucial role in maintaining host health. In
the context of this study, FMT modifies gut microbiota by transferring the diverse
microorganism community from mND or mHFD pregnant mice (Supplementary
Fig.6a), enabling the investigation of the impact of microbiota alterations on host
health. Pre-treatment with antibiotics for one week prior to FMT eradicates the
recipient's indigenous microbial community, creating a favorable environment for
successful establishment and colonization of the transplanted microorganisms. Our
findings indicate that an imbalance in the gut microbiota hinders the efficient
absorption of VB1 from the gut, consequently impacting the formation of primordial
follicles in offspring (see Supplementary Figure 6).

Supplementary Fig.6

**Figure Legends: Supplementary Fig.6 a** Study design of the fecal microbiota transplant
(FMT) experiment.

**Question 13: Figure 5h, i, image very small and not convincing.**

**Response:** Thank you for your suggestion. We have generated new images with
 improved clarity and resolution (Fig.5a), allowing for a more detailed and convincing
 representation of the data. These new images highlight specific regions of interest,
 such as the germ cells (DDX4 positive cells) and granulosa cells (flattened cells
 surrounding germ cells) (Niu & Spradling, 2020; J. J. Wang et al., 2020), and provide
 enhanced visualization of the expression of PDH-E1 in the cell nuclei (Fig.5b).

**Reference:**

Niu, W., & Spradling, A. C. (2020). Two distinct pathways of pregranulosa cell differentiation support
 follicle formation in the mouse ovary. *Proc Natl Acad Sci U S A*, 117(33), 20015-20026.
 doi:10.1073/pnas.2005570117
 Wang, J. J., Ge, W., Zhai, Q. Y., Liu, J. C., Sun, X. W., Liu, W. X., . . . Shen, W. (2020). Single-cell
 transcriptome landscape of ovarian cells during primordial follicle assembly in mice. *PLoS*
 *Biol*, 18(12), e3001025. doi:10.1371/journal.pbio.3001025

Fig.5

 **Figure Legends:** *Fig.5 a* IF staining of PDC-E1 and DDX4 in ovary. PDC-E1, DDX4 and
 DAPI are stained in magenta, green and blue, respectively. Scale bar, 50 µm. *b* Quantify the
 fluorescence intensity of PDC-E1, DDX4, and DAPI in granulosa and germ cells along the
 white dashed lines (*a*). The horizontal dashed gray lines (*b*) indicate the baseline fluorescence
 intensity, while the vertical dashed gray lines represent the demarcation between germ cells
 and granulosa cells.

**Question 14:** Figure 6m, n, how were granulosa cells identified also images are
 very small and not convincing.

**Response:** Thank you for your suggestion. In previous versions of the manuscript,

granulosa cells were identified based on their positional relationship with germ cells.
Specifically, the identification of granulosa cells relied on their distinct arrangement
as a single layer of flattened cells surrounding the germ cells, which encompassed
both nests and follicles (Niu & Spradling, 2020; J. J. Wang et al., 2020). As there were
no specific antibodies available for direct labeling of granulosa cells (Please refer to
the response to **Question 19** for details), we utilized DDX4 (magenta) as a marker for
germ cells, which allowed us to determine the presence of granulosa cells based on
their spatial association with the germ cell population.

In the revised manuscript, we have extensively optimized the experimental
protocol to determine the most optimal approach for labeling granulosa cells using
FOXL2 (Schmidt et al., 2004). In response to your comment, we have provided larger
and more visually informative images in the revised manuscript to better showcase the
identification granulosa cells (Fig.6k and Supplementary Fig.12e). Additionally, we
assessed granulosa cell proliferation by analyzing the presence of double-positive
cells for FOXL2 (magenta) and EdU (green) in each experimental group. The results
of this analysis, as presented in Fig. 6l and Supplementary Fig. 12f, support the trends
observed in our DDX4-EDU co-staining experiments, further validating the
consistency of our findings.

**Reference:**

- Niu, W., & Spradling, A. C. (2020). Two distinct pathways of pregranulosa cell differentiation support
follicle formation in the mouse ovary. *Proc Natl Acad Sci U S A*, *117*(33), 20015-20026.
doi:10.1073/pnas.2005570117
- Wang, J. J., Ge, W., Zhai, Q. Y., Liu, J. C., Sun, X. W., Liu, W. X., . . . Shen, W. (2020). Single-cell
transcriptome landscape of ovarian cells during primordial follicle assembly in mice. *PLoS*
*Biol*, *18*(12), e3001025. doi:10.1371/journal.pbio.3001025
- Schmidt, D., Ovitt, C. E., Anlag, K., Fehsenfeld, S., Gredsted, L., Treier, A. C., & Treier, M. (2004).
The murine winged-helix transcription factor Foxl2 is required for granulosa cell
differentiation and ovary maintenance. *Development*, *131*(4), 933-942. doi:10.1242/dev.00969

Supplementary Fig.12

**Figure Legends:** **Fig.6 k, l** IF staining of EdU (Proliferation labeling) and FOXL2 (Granular
 cell marker), and quantification of FOXL2-EdU double-positive cells per section in offspring
 ovaries from mND, mHFD, mTD or mHFD+VB1 mice, respectively ($n = 6$ biologically
 independent repeats). EdU, FOXL2, and DNA are stained in green, magenta, and blue,
 respectively. Scale bar, 50 μm . Statistical analyses were performed by one-way analysis of
 variance (ANOVA) with Tukey's test for multiple comparisons. **Supplementary Fig.12 e, f**
 staining of EdU (Proliferation labeling) and FOXL2 (Granular cell marker), and
 quantification of FOXL2-EdU double-positive cells ovarian section following treatment with
 CPI-613, Scr.siRNA or Pdha1-siRNA, respectively ($n = 6$ biologically independent repeats).
 EdU, FOXL2, and DNA are stained in green, magenta, and blue, respectively. Scale bar, 50
 297 μm . Student's *t* test (two-tailed) was used for statistical analysis.

**Question 15: How were participants identified for follow up and were they from**

**an ongoing study?**

**Response:** We greatly appreciate your professional review of our manuscript. The
participants included in the follow-up were identified among pregnant women
attending hospital appointments within one month. These participants were
prospectively recruited for the study and did not originate from an ongoing study.

Prior to participation, they provided informed consent. The inclusion criteria specified
pregnant women at approximately 26 weeks of gestation. Exclusion criteria
comprised pre-existing diabetes, metabolic disorders, prior antibiotic use, history of

alcohol or drug abuse, or the need for treatment for chronic diseases. Following
recruitment, all participants were screened for gestational diabetes mellitus by an oral
glucose tolerance test (Vinter et al., 2018). This approach ensured the inclusion of
both healthy individuals and those with gestational diabetes, allowing for comparative
analysis and investigation of specific outcomes related to the condition.

**Reference:**

Vinter, C. A., Tanvig, M. H., Christensen, M. H., Ovesen, P. G., Jorgensen, J. S., Andersen, M. S., . . .
Jensen, D. M. (2018). Lifestyle Intervention in Danish Obese Pregnant Women With Early
Gestational Diabetes Mellitus According to WHO 2013 Criteria Does Not Change Pregnancy
Outcomes: Results From the LiP (Lifestyle in Pregnancy) Study. *Diabetes Care*, *41*(10), 2079-
2085. doi:10.2337/dc18-0808

**Question 16: Line 320, define steatosis and how to identify in Figure 7d.**

**Response:** We appreciate your professional suggestion. We have provided an updated
image in Fig.7d to better illustrate this phenomenon. The updated image clearly shows
the presence of a large number of lipid droplets (black arrows) in the liver cells of the
mHFD group, which is one of the typical manifestation of liver steatosis (Friedman,
Neuschwander-Tetri, Rinella, & Sanyal, 2018; Powell, Wong, & Rinella, 2021;
Takahashi & Fukusato, 2014).

**Reference:**

Takahashi, Y., & Fukusato, T. (2014). Histopathology of nonalcoholic fatty liver disease/nonalcoholic
steatohepatitis. *World J Gastroenterol*, *20*(42), 15539-15548. doi:10.3748/wjg.v20.i42.15539
Friedman, S. L., Neuschwander-Tetri, B. A., Rinella, M., & Sanyal, A. J. (2018). Mechanisms of
NAFLD development and therapeutic strategies. *Nat Med*, *24*(7), 908-922.
doi:10.1038/s41591-018-0104-9
Powell, E. E., Wong, V. W., & Rinella, M. (2021). Non-alcoholic fatty liver disease. *Lancet*,
*397*(10290), 2212-2224. doi:10.1016/S0140-6736(20)32511-3

**Fig.7**

**Figure Legends: Fig.7 d Representative H&E images of liver in mND and mHFD mice.**

*Black arrows indicate lipid droplets. Scale bars, 50 μ m.*

**Question 17: Line 321, the levels of GLU and TG do not look similar in ND and**
**mHFD.**

**Response:** Thank you for bringing this issue to our attention. We have revised our
description in line 347, *“To test whether the maternal HFD model aligned with the*
*phenotypic similarities with GDM, we revealed that the hepatocytes containing lipid*
*droplets in the mHFD mice, as demonstrated by H&E staining (Fig.7d), and GLU and*
*TG levels were significantly elevated in the mHFD group compared with mND mice*
*(Fig.7e).”*.

**Question 18: Figure 1a, what are the 2 green rectangles?**

**Response:** We appreciate your professional comments. We clarify that the two green
rectangles represent a high-fat diet (D12492, Research Diets, Inc.). We apologize for
any confusion caused. In response, we have removed these two green rectangles in the
revised version to avoid any potential misunderstanding. In addition, we have
appropriately labeled the corresponding areas on the figure as “Normal diet or high-
fat diet”.

**Question 19: Line 1057, what is in red, why not use a granulosa cell marker?**

**Response:** Thank you for your suggestion. In previous versions of the manuscript, the
magenta color staining depicted the expression of DDX4, a widely recognized germ
cell marker. While the use of a granulosa cell-specific marker would have been ideal,
unfortunately, we encountered limitations in finding a suitable antibody that
specifically labels granulosa cells.

We attempted various antibodies of granulosa cell marker, and through
performing optimization of experimental procedures, we found that anti -FOXL2
(ab5096, abcam) exhibited notable staining positivity at a dilution ratio of 1:60 and
with an extended primary antibody staining time. As addressed in response to
question 14, we have incorporated a proliferation analysis using granulosa cell

markers in the revised manuscript. This analysis involved co-staining with EDU,
which allowed for clear visualization and quantification of the number of EDU-
positive cells within the FOXL2-labeled granulosa cell population (Fig. 6k and
Supplementary Fig. 12e). The statistical analysis results obtained were consistent with
the trends observed in our DDX4-EDU staining analysis (Fig. 6l and Supplementary
Fig. 12f).

**Reference:**

Niu, W., & Spradling, A. C. (2020). Two distinct pathways of pregranulosa cell differentiation support
follicle formation in the mouse ovary. *Proc Natl Acad Sci U S A*, *117*(33), 20015-20026.

doi:10.1073/pnas.2005570117

Wang, J. J., Ge, W., Zhai, Q. Y., Liu, J. C., Sun, X. W., Liu, W. X., . . . Shen, W. (2020). Single-cell
transcriptome landscape of ovarian cells during primordial follicle assembly in mice. *PLoS*

*Biol*, *18*(12), e3001025. doi:10.1371/journal.pbio.3001025

**Minor issues:**

**Question 1: The whole document needs to be checked for correct spelling and**
**grammar, some instances noted here.**

**Response:** Thank you for your suggestion. We have reviewed the document
thoroughly to correct errors of spelling and grammar.

**Question 2: Line 99, “examined by using immunostained” should be “examined**
**by immunostaining”.**

**Response:** We feel sorry for our carelessness. We have revised the manuscript and
made the suggested modification in line 99 “*examined by immunostaining*”. We
appreciate your attention to detail.

**Question 3: Line 128, the R package, Monocle (instead of just ‘monocle).**

**Response:** Thanks for your suggestion. We have made the revisions in our manuscript
in line 191, “*Subsequently, we extracted germ cells subpopulation and monocle were*
*performed in-depth analysis using R Package “Monocle” (Supplementary Fig.7e, f).*”.

**Question 4: Line 134-135, I think the wording is not correct---you didn't enrich**

**the genes?**

**Response:** Thank you for bringing this to our attention. According to your
suggestions, we have revised the sentence in line 196, *“To explore the genetic*
*characteristics of germ cells at state 2, Gene Ontology (GO) enrichment analysis was*
*performed to investigate the gene function categories (Fig.3d).”*.

**Question 5: Line 135-137, again the wording is not correct-“we gained insight**
**into the GO terms” ?**

**Response:** Thank you for bringing this to our attention. We have made the necessary
modifications in line 199: *“Obviously, the GO terms of ‘mitochondrion organization’,*
*‘mitochondrial respiratory chain complex assembly’, and ‘ATP metabolic process ’*
*were highly enriched in germ cells at state 2 (Fig.3d).”*.

**Question 6: Line 172, “attempted to design” either you designed it or you didn’t.**

**Response:** Thank you for your suggestion. We have made the necessary modification
in line 139, *“To investigate the potential function of VBI in primordial follicle*
*formation, we performed the experiment as shown in Fig.2e and Supplementary*
*Fig.4a.”*.

**Question 7: Line 177, 178, delete “the development”.**

**Response:** Thanks for your suggestion. We have made the necessary modifications to
remove the phrase “the development” from the sentence.

**Question 8: Line 205-207 does not make sense.**

**Response:** Thank you for your comment regarding lines 205-207 of our manuscript.
We have revised in line 171: *“These findings suggest that maternal HFD disrupts the*
*balance of gut microbiota, leading to impaired absorption of VBI during pregnancy.”*.

**Question 9: Line 270, do they really mean “lost”?**

**Response:** We appreciate your professional suggestion. In the original manuscript, we

used the term "lost" to refer to the down-regulated promoter regions in the context of
our analysis. However, upon further review and considering your suggestion, we have
revised in line 296 to better convey our intended meaning, "*These genes at the*
*promoter regions with reduced accessibility were predominantly enriched in the*
*'Protein processing in endoplasmic reticulum', 'Ribosome', and 'Cell cycle' pathways*
*(Fig.6d)."*

**Question 10: The references are separated, some are on page 22 and some on**
**page 39.**

**Response:** Thank you for bringing this to our attention. The purpose of this is to list
the references separately between the text and the Materials and Methods section,
numbered in the order in which they are cited in this section. To avoid reader
confusion, we ensure that all references are properly numbered and placed in a
continuous list of the revised manuscript.

**Question 11: Line 578, what is after breakfast fasting?**

**Response:** Thank you for your suggestion. What was intended to be conveyed is that
blood samples were collected from the study participants after an overnight fast of at
least 12 hours, which was done to standardize the basal metabolic state of the
participants prior to collecting the blood samples. We have revised it in Line 495:

"*Maternal blood samples were collected from participants following an overnight fast*
*of at least 12-h to standardize their basal metabolic state prior to sample collection.*"

**Question 12: Line 825 and 830 what are PG cells?**

**Response:** Thank you for your suggestion. We have made the necessary correction of
replacing "PG cells" with "granulosa cells". It is noteworthy that in current research,
some scholars refer to granulosa cells as pregranulosa cells or PG cells. In this
manuscript, we maintain consistency and use the term "granulosa cells" uniformly.

**Question 13: Figure 2a, bracoding should be barcoding.**

**Response:** We feel sorry for our carelessness. According to your suggestion, we have
corrected the “bracoding” into “barcoding” in Fig.3a.

**Question 14: Figure 7j, k, l, relative is spelled wrong.**

**Response:** Thank you for bringing this to our attention. We have changed the spelling
error to “relative” in the revised manuscript. In addition, we will carefully review the
entire text and ensure that all spelling errors are corrected in the revised version.

**Reviewer #2 (Remarks to the Author):**

**This is a very well written and comprehensive study, including noteworthy**
**results. The research question is novel and the potential implications of the**
**results are impactful. The study includes a multi-faceted methodological**
**approach, which is elegant and well designed. Overall, results support the**
**conclusions made throughout the manuscript. Yet, there are a few points that**
**require clarification:**

**Question 1: The authors results show that maternal high-fat intake during**
**pregnancy induces VB1 insufficiency which, in turn, causes a low number of**
**primordial follicles in the offspring. Did the authors test whether ovarian**
**function in the F1 or F2 generation? If data are not available, please discuss.**

**Response:** We greatly appreciate your valuable review of our manuscript. We have
extensively evaluated the ovarian function in the F1 generation, as described in detail
in line 104 of the manuscript: “*Moreover, there were fewer primordial follicles and*
*growing follicles in the ovaries of 3-week-old and 7-month-old offspring from mHFD*
*mice (Fig.1h and Supplementary Fig.1d), indicating that the defects of ovarian*
*reserves had a long-range effect. In addition, we performed the experiments using*
*ICR mice, and observe identical phenotypes over the same period of HFD feeding*
*(Supplementary Fig.1e-h). These data confirmed that primordial follicle formation*

was disrupted from mHFD offspring. In addition to the loss of primordial follicles,
oocyte quality and developmental potentiality were also compromised in the ovaries
of 3-week-old offspring from mHFD mice (Supplementary Fig.2). Specifically, the rate
of aberrant spindles and misaligned chromosomes was significantly increased in the
mHFD group (Supplementary Fig.2a, b). In addition, the abnormal proportion of 2-
cell embryos, 4-cell embryos, and blastocysts was significantly increased after in vitro
fertilization in the mHFD group (Supplementary Fig.2c, d).”. These results suggest
that maternal high-fat intake during pregnancy impairs ovarian function in F1
generation.

**Question 2: All figures are beautifully presented; yet, in many cases the authors**
**are comparing multiple groups. Do they use multiple comparisons? The p values**
**are highly significant, which makes me wonder whether adjustment for multiple**
**comparison were used.**

**Response:** We appreciate your professional suggestion. We have provided explicit
information about the statistical methods used in each figure legends and in the
statistical analysis section of the Materials and Methods. All data in this study were
analyzed using GraphPad Prism 8 analysis software. For comparing two sets of data at
the same time point (e.g., P0_mND vs. P0_mHFD), a two-tailed unpaired Student's t-
test was performed. When evaluating datasets with three or more groups at the same
time point (e.g., mND, mHFD, mTD, mHFD+VB1), a one-way analysis of variance
(ANOVA) was conducted, followed by Tukey's test for multiple comparisons. The *P*-
value less than 0.05 was considered statistically significant, while the *P*-value less
than 0.01 indicated highly significant difference. In addition, this statistical analysis
approach has been widely employed in a multitude of studies (X. Liu et al., 2021;
Zeng et al., 2020; Zhang et al., 2021), highlighting its broad applicability in data
analysis.

**Reference:**

Liu, X., Li, X., Xia, B., Jin, X., Zou, Q., Zeng, Z., . . . Liu, X. (2021). High-fiber diet mitigates
maternal obesity-induced cognitive and social dysfunction in the offspring via gut-brain axis.

*Cell Metab*, 33(5), 923-938 e926. doi:10.1016/j.cmet.2021.02.002
Zeng, S. L., Li, S. Z., Xiao, P. T., Cai, Y. Y., Chu, C., Chen, B. Z., . . . Liu, E. H. (2020). Citrus
polymethoxyflavones attenuate metabolic syndrome by regulating gut microbiome and amino
acid metabolism. *Sci Adv*, 6(1), eaax6208. doi:10.1126/sciadv.aax6208
Zhang, T., Sun, P., Geng, Q., Fan, H., Gong, Y., Hu, Y., . . . Zhou, Y. (2022). Disrupted spermatogenesis
in a metabolic syndrome model: the role of vitamin A metabolism in the gut-testis axis. *Gut*,
71(1), 78-87. doi:10.1136/gutjnl-2020-323347

**Question 3: The fecal microbiota transplantation experiments require further**
**description. Can you please elaborate in the study groups and controls? Were**
**pregnant mice (without HFD) transplanted with fetal microbiota of dams with**
**HFD and vice versa? This is not clear in Figure 4H. Also, what are the controls?**

**Response:** Thanks for your professional review work on our manuscript. According to
your comment, we have made necessary revisions to the FMT schematic to ensure an
accurate representation of the experimental design (Supplementary Fig.6a). In this
study, pregnant mice (without HFD) were not transplanted with the fetal microbiota of
dams with HFD, and vice versa. Pre-treatment with antibiotics for one week prior to
FMT eradicates the recipient's indigenous microbial community, creating a favorable
environment for successful establishment and colonization of the transplanted
microorganisms. Two groups were defined: the control group (mND-FMT) and the
study group (mHFD-FMT). The control group (mND-FMT): normal diet recipient
mice (pregnant mice) transplanted with fecal microbiota from the normal diet
pregnant mice. This group serves as a control to assess the baseline effects of the
mND microbiota on the recipients. The study group (mHFD-FMT): normal diet
recipient mice (pregnant mice) transplanted with fecal microbiota from the high-fat
diet pregnant mice. This group is designed to investigate the specific effects of the
HFD-induced microbiota on the recipients.

Supplementary Fig.6

**Figure Legends: Supplementary Fig.6 a** Study design of the fecal microbiota transplant
(FMT) experiment.

**Question 4: Maternal age was significantly different between GMD and HC**
**women. Did the authors adjusted for maternal age?**

**Response:** Thank you for your suggestion. Despite the significantly difference in age
between GMD and HC women, we did not adjust for maternal age in our analysis.

To address the potential influence of maternal age on our findings, we performed
a multiple linear regression analysis. The dependent variable was the area under the
curve (AUC) of the oral glucose-tolerance test (oGTT), while the independent
variables included relative VB1 levels and maternal age. We specifically examined the
association between relative VB1 levels and oGTT (AUC) while controlling for age to
mitigate issues such as multicollinearity. The results of our analysis, as depicted in
Figure a ($P=0.11$), indicated that there was no significant association between relative
VB1 levels and maternal age, thereby minimizing the risk of multicollinearity.

Subsequently, we derived a multiple linear regression equation, as shown in Figure b,
which yielded the following relationship: “oGTT (AUC) = 12.389 – 0.063*(Relative
VB1 levels, $P=0.002$) + 0.208*(Age, $P=0.013$)”. These findings suggest that the
observed associations significantly between maternal VB1 levels and gestational
diabetes were not influenced by maternal age.

Nevertheless, we had reservations about including age as a covariate due to the
potential for added complexity and confounding effects. Furthermore, considering the

multifaceted nature of the study population, it raised questions about whether other
 factors such as maternal height, weight, BMI, and blood pressure should also be taken
 into account. As a result, we decided not to adjust for maternal age in our analysis, as
 we deemed it prudent to maintain a more focused examination of the specific
 variables under investigation.

 **Figure Legends:** *a* Spearman's correlation analysis between vitamin B1 and age in HC and
 GDM. *b* The table displays the multivariate linear regression equation for the dependent
 variable, which is the area under the curve (AUC) of the oral glucose tolerance test (oGTT).
 The independent variables in the equation include the relative level of VB1 and maternal age.

**Question 5: In general, the manuscript is beautifully written and the results are**
 **presented in a very compelling manner.**

**Response:** We greatly appreciate the reviewer's positive evaluation on our manuscript.

**Reviewer #3 (Remarks to the Author):**

**This study intends to uncover the role of vitamin B1 deficiency in dam gut**
 **attributing to growth failure of primordial follicle in maternal HFD-induced**
 **fetuses, the major findings include: 1) identified pregnant HFD induces dam and**

fetal serum vitamin B1 deficiency, which is associated with dam microbiome
disruption; 2) vitamin B1 deficiency inhibits ovary PDH activity, the decrease of
acetyl-CoA further impairs histone acetylation of proliferative genes in granulosa
cells; 3) gestational diabetes also decrease serum vitamin B1 in dams.
The manuscript is readable and the study is interesting, but there are several

**major concerns required to address:**

**Question 1: The pregnant HFD impacts on fetal ovary development should be**
**due to multidimensional factors, such as excessive of saturated and unsaturated**
**fatty acids intake, thus it is vital to block vitamin B1 receptor in fetal follicular to**
**determine the actual impacts of vitamin B1 deficiency on primordial follicle**
**maturation and gene acetylation.**

**Response:** Thank you very much for your valuable suggestion. We have provided
additional experiments regarding the block of vitamin B1 transporter in the ovaries of
fetuses. It is important to note that vitamin B1 does not exert its effects through direct
receptor binding but rather through two transport proteins, THTR1 and THTR2, the
products of the *Slc19a2* and *Slc19a3* genes, respectively (Ganapathy, Smith, & Prasad,
2004; Uebanso, Shimohata, Mawatari, & Takahashi, 2020; Zhao & Goldman, 2013).
We considered using RNA interference to decrease the expression of *Slc19a2/Slc19a3*
and block the vitamin B1 transporter in the ovaries of fetuses. The results
demonstrated that *Slc19a2* siRNA-1547 and *Slc19a3* siRNA-236 effectively inhibited
the protein expression of SLC19A2 and SLC19A3, respectively (Supplementary
Fig.9a, b). Interestingly, inhibiting SLC19A2 led to an upregulation of SLC19A3,
while inhibiting SLC19A3 did not affect SLC19A2 expression (Supplementary Fig.9
c, d), which means there is a compensatory mechanism between these transport
proteins. To effectively block the vitamin B1 transporter, we co-transfected *Slc19a2*
siRNA-1547 and *Slc19a3* siRNA-236, resulting in a simultaneous decrease in
SLC19A2/SLC19A3 expression and a significant reduction in the acetyl-CoA content
in ovarian tissue (Fig.4i and Supplementary Fig.9e, f).

Furthermore, amprolium (APL) was also found to block the utilization of vitamin
B1 in cell line culture (Bunik, Tylicki, & Lukashev, 2013). Therefore, we employed
500 μ M and 1 mM APL *in vitro* to inhibit vitamin B1 utilization, resulting in a
significant decrease in acetyl-CoA content at 1 mM APL (Fig.4j). Importantly, both
the *Slc19a2/Slc19a3* siRNA group and the APL group exhibited significant inhibition
of primordial follicle formation compared to their respective control groups, although

the total number of follicles remained unchanged (Fig.4k-m). Additionally, the
 detection of histone acetylation levels revealed that both the *Slc19a2/Slc19a3* siRNA
 group and the APL group exhibited significant reductions in AC-H3 and AC-H4 levels
 compared to the control group (Fig.5g, h). These findings suggest that blocking the
 utilization of vitamin B1 in fetal ovaries significantly affects the levels of histone
 acetylation and primordial follicle formation.

**Reference:**

Ganapathy, V., Smith, S. B., & Prasad, P. D. (2004). SLC19: the folate/thiamine transporter family.
 *Pflugers Arch*, 447(5), 641-646. doi:10.1007/s00424-003-1068-1
 Uebanso, T., Shimohata, T., Mawatari, K., & Takahashi, A. (2020). Functional Roles of B-Vitamins in
 the Gut and Gut Microbiome. *Mol Nutr Food Res*, 64(18), e2000426.
 doi:10.1002/mnfr.202000426
 Zhao, R., & Goldman, I. D. (2013). Folate and thiamine transporters mediated by facilitative carriers
 (SLC19A1-3 and SLC46A1) and folate receptors. *Mol Aspects Med*, 34(2-3), 373-385.
 doi:10.1016/j.mam.2012.07.006
 Bunik, V. I., Tylicki, A., & Lukashev, N. V. (2013). Thiamin diphosphate-dependent enzymes: from
 enzymology to metabolic regulation, drug design and disease models. *FEBS J*, 280(24), 6412-
 6442. doi:10.1111/febs.12512

Supplementary Fig.9

**Figure Legends: Supplementary Fig.9 Optimization of SLC19A2 and SLC19A3 for**
 **interference efficiency screening. a, b** Relative protein levels of SLC19A2 and SLC19A3 after
 ovarian transfected with *Slc19a2*-siRNA for 6-h and cultured for 72-h. GAPDH were loading
 controls ($n = 3$ biologically independent repeats). **c, d** Relative protein levels of SLC19A3 and
 SLC19A2 after ovarian transfected with *Slc19a3*-siRNA for 6-h and cultured for 72-h.
 GAPDH were loading controls ($n = 3$ biologically independent repeats). **e, f** Relative protein
 levels of SLC19A2 and SLC19A3 in ovaries treated with *Sci*-siRNA or *Slc19a2*-
 *siRNA*+*Slc19a3*-*siRNA*, respectively ($n = 3$ biologically independent repeats). Uncropped
 blots in Source Data.

**Figure Legends: Fig.4 i** Relative Acetyl-CoA levels of *Scr*.*siRNA*-treated or *Slc19a2*-
 *siRNA*+*Slc19a3*-*siRNA*-treated ovaries. **j** Relative Acetyl-CoA levels of Control-treated, 500
 μM or 1mM amprolium (APL)-treated ovaries mice. **k** IF staining of DDX4 in ovaries treated

with 1mM APL, *Sci-siRNA* or *Slc19a2-siRNA+Slc19a3-siRNA*, respectively. *DDX4* and *DNA*
are stained in magenta and dark indigo, respectively. Scale bar, 50 μm . **l** The percentages of
oocytes within follicles in ovaries treated with 1mM APL, *Sci-siRNA* or *Slc19a2-*
*siRNA+Slc19a3-siRNA*, respectively ($n = 6$ biologically independent repeats). **m** The total
number of oocytes per section in ovaries treated with 1mM APL, *Sci-siRNA* or *Slc19a2-*
*siRNA+Slc19a3-siRNA*, respectively ($n = 6$ biologically independent repeats).

**Fig.5 g, h** Representative image and relative protein levels of *Ac-H3* and *Ac-H4* in ovaries
treated with 1mM APL, *Sci-siRNA* or *Slc19a2-siRNA+Slc19a3-siRNA*, respectively. *H3* and
*H4* were loading controls ($n = 3$ biologically independent repeats). Uncropped blots in
Source Data.

**Question 2: It is not convincing how gestational diabetes are able to link to**
**pathology of pregnancy HFD feeding.**

**Response:** Thank you for your suggestion. In our study, we aimed to explore the
adverse effects of HFD feeding during pregnancy and hope it would give instructive
meaning in the clinical research. Despite a significant increase in body weight in
pregnant mice of HFD group compared with ND group was observed, it did not meet
the criteria for obesity based on weight gain (de Moura et al., 2021). However, the
HFD intake during pregnancy led to insulin resistance and glucose intolerance in mice,
which is more consistent with the diagnostic criteria for gestational diabetes in
humans (American Diabetes Association Professional Practice, 2022). This
resemblance prompted us to recruit gestational diabetes mellitus (GDM) subjects to
further investigate whether the significantly decreased metabolite VB1, which we
found in the mHFD mice model would alter in the GDM.

**Reference:**

de Moura, E. D. M., Dos Reis, S. A., da Conceicao, L. L., Sedyama, C., Pereira, S. S., de Oliveira, L.
681 L., . . . Milagro, F. I. (2021). Diet-induced obesity in animal models: points to consider and
682 influence on metabolic markers. *Diabetol Metab Syndr*, 13(1), 32. doi:10.1186/s13098-021-
683 00647-2
American Diabetes Association Professional Practice, C. (2022). 2. Classification and Diagnosis of
Diabetes: Standards of Medical Care in Diabetes-2022. *Diabetes Care*, 45(Suppl 1), S17-S38.
doi:10.2337/dc22-S002

**Question 3: The author examined 16S ribosomal DNA (rDNA) amplicon**
**sequencing, and find that the 43 bacteria were altered at the genus level in the**
**mHFD group compared with the mND group. Further, the author mentioned**
**that these findings above prompted us to consider that these microbiotas**
**appeared to contribute to VB1 metabolism and absorption. However, the author**
**did not use metagenomics methods to check whether or not these microbiome**
**can synthesize vitamin, therefore, the author should explore the mechanism**
**underlying the VB1 synthesis by gut microbe.**

**Response:** We sincerely appreciate the thorough review provided by the reviewer. It
is well known that vitamin B1, as an essential nutrient, is primarily absorbed through
the intestine. The main source of vitamin B1 in the body is from food intake, while
the synthesis by gut microbiota contributes approximately 2.3% (Magnusdottir,
Ravcheev, de Crecy-Lagard, & Thiele, 2015; Uebanso et al., 2020). Therefore,
changes in microbiota-mediated vitamin B1 synthesis have minimal impact on overall
vitamin B1 levels in the body. Furthermore, it is revealed that the absorption of
vitamin B1 from food is mainly through the SLC19A3, which is a transport protein
located in the outer membrane of the small intestine (Uebanso et al., 2020; Zhao &
Goldman, 2013), while vitamin B1 synthesis by gut microbiota is mainly occurs in the
colon (Nabokina et al., 2017). Hence, in our study, we focus on assessing the
expression levels of SLC19A3 transporter in the jejunum and ileum. Remarkably, we
observed a notable downregulation of SLC19A3 expression (Supplementary Fig.5i in
the revised manuscript), which was concomitant with compromised vitamin B1
absorption (Uebanso et al., 2020; Zhao & Goldman, 2013). However, the mechanisms
by which gut microbiota affects the expression of vitamin B1 transport protein still
need further investigation. In the revised version of the manuscript, we have rephrased
the sentence in line 158 as follows: *“Given that the main source of VB1 in the body is*
*from food intake, we hypothesis whether the dysbiosis of micro-ecological balance*
*may disrupt VB1 absorption.”.*

Although the vitamin B1 synthesis by gut microbiota in the host is relatively

minor, we addressed the reviewer's comment and examined the gut microbiota with
the ability to synthesize vitamin B1. Our study employed third-generation full-length
16s rDNA sequencing, which differs from traditional approaches that target single or a
few variable regions (Mosher, Bernberg, Shevchenko, Kan, & Kaplan, 2013). This
advanced sequencing method allows for the detection of complete microbial
sequences, significantly enhancing the resolution at the species level and providing a
more comprehensive analysis of the microbial composition and their potential for
vitamin B1 synthesis (Sadowsky et al., 2017). Hence, we did not conduct additional
metagenomic sequencing but instead utilized the PICRUSt2 package to perform
functional predictions based on the 16s rDNA sequencing results (Douglas et al.,
2020). It is noteworthy that the PICRUSt2 package is also utilized for functional
predictions of microbial communities based on metagenomic data (Douglas et al.,
2020).

As shown in Figure a, the PICRUSt2 algorithm utilizes amplicon sequence
variants (ASVs) and their abundances as the basis for functional predictions, resulting
in more accurate estimations of gene family and pathway abundances. Through our
analysis of the 16S rDNA sequencing results, we found the top 5 genus level bacteria
associated with thiamine (vitamin B1) metabolism (Figure b). Cheng et al. conducted
a study that identified nine bacterial genera significantly associated with thiamine
(vitamin B1) synthesis (Chen, Wang, et al., 2021), of which three genera were found
to be shared with our study (Figure c). Among them, *Lachnoclostridium* and
*Bacteroides* showed a significant increase in abundance in the mHFD group compared
to the mND group, while the abundance of *Lachnospiraceae_NK4A136_group* did not
exhibit a significant change (Figure d). Further analysis revealed that except for these
three genera, seven bacterial species were significantly enriched; however, these
seven species did not show significant alterations between the two groups (Figure e).
In summary, our functional analysis identified three candidate genera that may be
significantly associated with VB1synthesis, and these genera include seven candidate
species. This information can help us understand the potential contributions of
specific bacterial groups to VB1 metabolism in the gut microbiota.

Figure Legends: **a** The PICRUSt2 method consists of phylogenetic placement, hidden-state prediction and sample-wise gene and pathway abundance tabulation. ASV sequences and abundances are taken as input, and gene family and pathway abundances are output. All necessary reference tree and trait databases for the default workflow are included in the PICRUSt2 implementation. **b** Bacteria at the top 5 generic levels associated with thiamine (vitamin B1) metabolism. **c** Venn diagram illustrating the relationship of the bacterial genera between our data and Cheng et al. **d** Comparison of the relative abundance of three genera. **e** Comparison of the relative abundance of seven species. The two-tailed student's *t* test was used for statistical analysis.

**Reference:**

Magnusdottir, S., Ravcheev, D., de Crecy-Lagard, V., & Thiele, I. (2015). Systematic genome assessment of B-vitamin biosynthesis suggests co-operation among gut microbes. *Front Genet*, 6, 148. doi:10.3389/fgene.2015.00148

Uebanso, T., Shimohata, T., Mawatari, K., & Takahashi, A. (2020). Functional Roles of B-Vitamins in the Gut and Gut Microbiome. *Mol Nutr Food Res*, 64(18), e2000426.

doi:10.1002/mnfr.202000426
Zhao, R., & Goldman, I. D. (2013). Folate and thiamine transporters mediated by facilitative carriers
(SLC19A1-3 and SLC46A1) and folate receptors. *Mol Aspects Med*, 34(2-3), 373-385.
doi:10.1016/j.mam.2012.07.006
Nabokina, S. M., Inoue, K., Subramanian, V. S., Valle, J. E., Yuasa, H., & Said, H. M. (2017).
Molecular identification and functional characterization of the human colonic thiamine
pyrophosphate transporter. *J Biol Chem*, 292(40), 16526. doi:10.1074/jbc.A113.528257
Mosher, J. J., Bernberg, E. L., Shevchenko, O., Kan, J., & Kaplan, L. A. (2013). Efficacy of a 3rd
generation high-throughput sequencing platform for analyses of 16S rRNA genes from
environmental samples. *J Microbiol Methods*, 95(2), 175-181.
doi:10.1016/j.mimet.2013.08.009
Sadowsky, M. J., Staley, C., Heiner, C., Hall, R., Kelly, C. R., Brandt, L., & Khoruts, A. (2017).
Analysis of gut microbiota - An ever changing landscape. *Gut Microbes*, 8(3), 268-275.
doi:10.1080/19490976.2016.1277313
Douglas, G. M., Maffei, V. J., Zaneveld, J. R., Yurgel, S. N., Brown, J. R., Taylor, C. M., . . . Langille,
778 M. G. I. (2020). PICRUSt2 for prediction of metagenome functions. *Nat Biotechnol*, 38(6),
685-688. doi:10.1038/s41587-020-0548-6
Chen, L., Wang, D., Garmaeva, S., Kurilshikov, A., Vich Vila, A., Gacesa, R., . . . Fu, J. (2021). The
long-term genetic stability and individual specificity of the human gut microbiome. *Cell*,
184(9), 2302-2315 e2312. doi:10.1016/j.cell.2021.03.024

**Question 4: The fecal microbiota transplantation (FMT) increased the levels of**
**VB1 and the protein expression levels of SLC19A3 was decreased in intestinal**
**digesta from mHFD recipient mice (mHFD-FMT) compared with mND recipient**
**mice (mND-FMT), respectively (Fig.4i, j). I just wonder that which**
**microorganisms plays a key role in the synthesis of VB1, so, more experiments**
**are needed to test in this section.**

**Response:** Thank you for the valuable suggestions. Through the fecal microbiota
transplantation (FMT), we have demonstrated that changes in the gut microbiota can
indeed influence the expression of SLC19A3 in intestinal cells (Supplementary Fig.6).
This dysregulation of SLC19A3 hinders the efficient absorption of VB1 from dietary
sources, resulting in inadequate serum levels of VB1 and accumulation of VB1 in the
intestinal digesta (Supplementary Fig.6). As previously stated in our response to
**Question 3**, the microbial synthesis of VB1 makes a modest contribution,
approximately 2.3%, to the total body requirement of VB1 (Magnusdottir et al., 2015;
Uebanso et al., 2020). Furthermore, the absorption of microbiota-synthesized VB1

predominantly takes place in the colon (Nabokina et al., 2017). However, it should be
noted that the primary focus of our study does not revolve around the microbiota's
synthesis of vitamin B1, but gut microbiota dysbiosis impact on overall serum vitamin
B1 levels.

Despite this, we utilized the PICRUST2 package to perform more accurate
predictions of potential microbiota involved in VB1 synthesis (**Question 3:** Figure a).
Through a comparative analysis with Chen's study (Chen, Wang, et al., 2021), we
identified three candidate genera that may be significantly associated with VB1
synthesis, and these genera include seven candidate species (**Question 3:** Figure b-e).

**Reference:**

Magnusdottir, S., Ravcheev, D., de Crecy-Lagard, V., & Thiele, I. (2015). Systematic genome
assessment of B-vitamin biosynthesis suggests co-operation among gut microbes. *Front Genet*,
6, 148. doi:10.3389/fgene.2015.00148

Uebanso, T., Shimohata, T., Mawatari, K., & Takahashi, A. (2020). Functional Roles of B-Vitamins in
the Gut and Gut Microbiome. *Mol Nutr Food Res*, 64(18), e2000426.
doi:10.1002/mnfr.202000426

Nabokina, S. M., Inoue, K., Subramanian, V. S., Valle, J. E., Yuasa, H., & Said, H. M. (2017).
Molecular identification and functional characterization of the human colonic thiamine
pyrophosphate transporter. *J Biol Chem*, 292(40), 16526. doi:10.1074/jbc.A113.528257

Chen, L., Wang, D., Garmaeva, S., Kurilshikov, A., Vich Vila, A., Gacesa, R., . . . Fu, J. (2021). The
long-term genetic stability and individual specificity of the human gut microbiome. *Cell*,
184(9), 2302-2315 e2312. doi:10.1016/j.cell.2021.03.024

**Reviewer #4 (Remarks to the Author):**

**This manuscript describes the potential role for reduced thiamine levels as a**
**mediator of alterations in oocyte development in female offspring of female mice**
**fed a HFD upon initiation of pregnancy. The authors demonstrate using**
**microbiome analysis and fecal transplantation that maternal HFD induces a**
**change in the microbiome, associated with reduced expression of the thiamine**
**transporter in maternal intestine and parallel changes in thiamine levels in the**
**maternal circulation. These changes are associated with altered developmental**
**trajectories of the developing fetal germ cells and expression of nuclear-encoded**

**mitochondrial genes and mitochondrial morphology. Modulation of thiamine**
**levels and/or signaling reversed these effects, which were also associated with**
**parallel changes in post-translational histone acetylation and chromatin**
**accessibility. Finally, the authors present data that thiamine levels are reduced in**
**humans with gestational DM.**

**Major issues:**

**Question 1: In general, the manuscript is interesting and the reversal**
**experiments are quite striking. However, the manuscript is challenging to read as**
**results are not clearly described. Limitations of the study are not well-delineated.**
**Moreover, the human data are not well-linked to the rest of the paper. More**
**information is needed about the women with GDM. Were they obese as**
**compared with the controls? Matched for age? Why not just study a cohort of**
**women with obesity since that is what the rest of the manuscript is focused.**

**Response:** We appreciate your suggestions. We have revised the manuscript to ensure
that the data is presented in a clear and concise manner.

Furthermore, our manuscript provides a discussion of the limitations of this study,
in line 386 of the revised manuscript: “*Moreover, further studies are needed to explore*
*the mechanisms by which gut microbiota affects the expression of vitamin B1*
*transport protein.*”; as well as in line 442 of the revised manuscript: “*Similarly, we*
*also observed that VBI supplementation was effective in preventing the maternal*
*HFD induction of low birth weight in offspring, although the precise underlying*
*mechanisms of this protective effect remain unclear. In addition, further exploration is*
*needed as to whether glucose tolerance can be improved by VBI supplementation in*
*mHFD group.*”.

Regarding the human data, we apologize for not providing enough information
about the participants. Following the reviewer' recommendations, we have
incorporated the demographic characteristics of women with GDM in supplementary
Table 1, encompassing their age, BMI and other pertinent information.

In addition, despite a significant increase in body weight in pregnant mice of

HFD group compared with ND group was observed, it did not meet the criteria for
 obesity based on weight gain (de Moura et al., 2021). However, the HFD intake
 during pregnancy led to insulin resistance and glucose intolerance in mice, which is
 more consistent with the diagnostic criteria for gestational diabetes (American
 Diabetes Association Professional Practice, 2022). Therefore, we investigated the
 participants with gestational diabetes in our study rather than obese individuals.

**Reference:**

de Moura, E. D. M., Dos Reis, S. A., da Conceicao, L. L., Sedyama, C., Pereira, S. S., de Oliveira, L.
 870 L., . . . Milagro, F. I. (2021). Diet-induced obesity in animal models: points to consider and
 871 influence on metabolic markers. *Diabetol Metab Syndr*, 13(1), 32. doi:10.1186/s13098-021-
 872 00647-2
 American Diabetes Association Professional Practice, C. (2022). 2. Classification and Diagnosis of
 Diabetes: Standards of Medical Care in Diabetes-2022. *Diabetes Care*, 45(Suppl 1), S17-S38.
 doi:10.2337/dc22-S002

**Question 2: Were levels of butyrate or acetate changed in the maternal**
 **circulation in the mice?**

**Response:** Thank you for your suggestions. We have performed additional testing
 using gas chromatography to measure the levels of short-chain fatty acids (SCFAs) in
 the maternal serum of mice. However, our study did not reveal any significant
 differences in the levels of SCFAs, including butyrate and acetate, between mice fed a
 high-fat diet and those fed a normal diet during pregnancy.

**Figure Legends: Figure** The serum concentrations of SCFAs in mND and mHFD groups. The
two-tailed student's *t* test was used for statistical analysis.

**Question 3: Please provide information about weight gain during pregnancy,**
**litter size, birth weight in the mice, including chow, HFD, thiamine deficiency,**
**vitamin B1 antagonist compound treatment, and B1 supplementation groups.**

**Response:** Thank you for your insightful comments on our manuscript. To provide a
comprehensive understanding of these outcomes, we collected data on weight gain
during pregnancy by monitoring the body weight of the pregnant mice at regular
intervals throughout gestation. Litter size was determined by counting the number of
pups born to each dam. Additionally, we measured the birth weight of the offspring
immediately after delivery. Statistical analysis revealed that at 19.5 days of gestation,
the mTA and mTD groups did not exhibit significant alterations in body weight
compared to the control group (Figure a and b). Notably, VB1 supplementation did
not significantly impact the body weight of pregnant individuals in the mHFD group
(Figure a and b). Moreover, there were no statistically significant differences observed
in litter size among the experimental groups (Figure c). Of particular significance,
newborn mice in the mHFD, mTA, and mTD groups displayed significantly reduced
body weight when compared to the mND group, whereas the VB1 supplement group
exhibited a significant recovery in body weight (Figure d).

In addition, we have made the decision not to include this information in the
revised manuscript. We believe that presenting these data would deviate from the
primary focus of our study and potentially compromise the clarity and coherence of
the manuscript.

**Figure Legends:** **Figure a** Body weight of pregnancy mice from mND, mHFD, mTA, mTD,
 and mHFD+VB1 groups (n = 5 mice for each group). **b** The weight of 19.5 days pregnant
 mice from mND, mHFD, mTA, mTD, and mHFD+VB1 groups (n = 10 mice for each group). **c**,
 **d** In the indicated groups, quantification of litter size and neonatal mouse body weight (n = 5
 litters for each group). Statistical analyses were performed by one-way analysis of variance
 (ANOVA) with Tukey's test for multiple comparisons.

**Question 4: For each of the omics analysis please provide information about**
 **whether data were corrected for multiple comparisons, and which method was**
 **used to do so?**

**Response:** We appreciate your professional suggestion. As stated in line 526 of the
 Materials and Methods section of the manuscript, the differential gene expression
 analysis was conducted for the single-cell RNA sequencing data utilizing the R
 package "Desingle". The corrected p-values in Desingle are typically calculated using
 the false discovery rate (FDR) correction method (Miao, Deng, Wang, & Zhang,
 2018). The FDR correction method, such as the Benjamini-Hochberg procedure,
 adjusts the P-values to control the expected proportion of false positives among the
 declared significant genes.

For the serum metabolomics analysis, 16S rDNA amplicon sequencing analyses,
 and ATAC-seq analysis in this study, the statistical method of Student's t-test was
 applied to assess the significance of observed differences. It is important to note that
 the use of Student's t-test for these omics analyses is widely accepted in the scientific
 community (Chen, Zhang, et al., 2021; X. Liu et al., 2021; Zhang et al., 2022).

**Reference:**

Miao, Z., Deng, K., Wang, X., & Zhang, X. (2018). DEsingle for detecting three types of differential
expression in single-cell RNA-seq data. *Bioinformatics*, 34(18), 3223-3224.
doi:10.1093/bioinformatics/bty332
Chen, L., Zhang, J., Zou, Y., Wang, F., Li, J., Sun, F., . . . Wang, C. Y. (2021). Kdm2a deficiency in
macrophages enhances thermogenesis to protect mice against HFD-induced obesity by
enhancing H3K36me2 at the Pparg locus. *Cell Death Differ*, 28(6), 1880-1899.
doi:10.1038/s41418-020-00714-7
Liu, X., Li, X., Xia, B., Jin, X., Zou, Q., Zeng, Z., . . . Liu, X. (2021). High-fiber diet mitigates
maternal obesity-induced cognitive and social dysfunction in the offspring via gut-brain axis.
*Cell Metab*, 33(5), 923-938 e926. doi:10.1016/j.cmet.2021.02.002
Zhang, T., Sun, P., Geng, Q., Fan, H., Gong, Y., Hu, Y., . . . Zhou, Y. (2022). Disrupted spermatogenesis
in a metabolic syndrome model: the role of vitamin A metabolism in the gut-testis axis. *Gut*,
71(1), 78-87. doi:10.1136/gutjnl-2020-323347

**Question 5: Figure 3D – Please color by group. It appears that there is not a**
**spectrum of B1 levels but rather stratification by group, in which case the**
**correlation has a someone different conclusion.**

**Response:** Thank you for your valuable comments. We would like to clarify that the
x-axis in Fig.3D (Supplementary Fig.3d in the revised version) represents a spectrum
of vitamin B1 levels. We generated separate correlation plots between vitamin B1
levels and follicles/nests for the P0 and P3 groups. In fact, we take the group
information into account when mapping correlations, although it is not visually
represented in the figure. However, in response to the reviewer's suggestion, we have
incorporated color-coded stratification of the groups in Supplementary Fig.3d of the
revised version to prevent potential misinterpretation by readers. We can confirm that
grouping the data did not affect the correlation values between the groups.

**Figure Legends: Supplementary Fig.3 d** Spearman's correlation analysis of vitamin B1

abundance with the proportion of oocytes within nests or within follicles.

Supplementary Fig.3

**Minor issues:**

**Question 1: Line 662 – what is “gray value of target proteins”?**

**Response:** Thanks for your kind comments. The “gray value of target proteins” refers
to the quantitative measurement of the intensity of the bands corresponding to a target
protein. The gray value is a numerical value that represents the intensity of the band,
with higher values indicating higher protein expression or abundance. This method of
protein quantification based on grayscale intensity is widely used (Guo et al., 2016; W.
X. Liu et al., 2022; Thomas et al., 2018) because it provides a quantitative measure of
protein expression and allows for comparisons between different experimental
conditions or samples.

**Reference:**

Guo, Y., Luo, W., Hu, Z., Li, J., Li, X., Cao, H., . . . Luo, D. (2016). Low expression of Aldo-keto
reductase 1B10 is a novel independent prognostic indicator for nasopharyngeal carcinoma.
*Cell Biosci*, 6, 18. doi:10.1186/s13578-016-0082-x
Liu, W. X., Tan, S. J., Wang, Y. F., Zhang, F. L., Feng, Y. Q., Ge, W., . . . Cheng, S. F. (2022). Melatonin
promotes the proliferation of primordial germ cell-like cells derived from porcine skin-derived
stem cells: A mechanistic analysis. *J Pineal Res*, 73(4), e12833. doi:10.1111/jpi.12833
Thomas, A. M., Ostroumov, A., Kimmey, B. A., Taormina, M. B., Holden, W. M., Kim, K., . . . Dani, J.
982 A. (2018). Adolescent Nicotine Exposure Alters GABA(A) Receptor Signaling in the Ventral
Tegmental Area and Increases Adult Ethanol Self-Administration. *Cell Rep*, 23(1), 68-77.
doi:10.1016/j.celrep.2018.03.030

**Question 2: Line 840 – what does “conforming to the standard of cell survival
rate” mean?**

**Response:** We feel great thanks for your professional review work on our manuscript.
For ATAC-seq, it is important to obtain high-quality chromatin accessibility data,
which requires a sufficient number of viable cells. Therefore, a minimum threshold
for the number of live cells or viability rate is often set to ensure the reproducibility
and accuracy of the sequencing results. This is because the quality and quantity of
chromatin accessible regions can be affected by factors such as cell death, chromatin
fragmentation, and DNA damage, which can lead to biases and noise in the data.

In order to enhance the clarity of the text and facilitate comprehension for the
reader, we have revised the sentence on line 762: “*Finally, the non-adherent germ*

*cells and ovarian tissue were removed to obtain granulosa cell suspension that met*
*sequencing criteria for cell survival rate.”.*

**Question 3: Line 879 – $p < 0.01$ should not be considered “extremely significant**
**difference.”**

**Response:** Thank you for your thorough review. We have revised the description in
line 803 “highly” instead of “extremely”.

REVIEWER COMMENTS

Reviewer #1 (Remarks to the Author):

the authors have sufficiently addressed all my concerns.

Reviewer #2 (Remarks to the Author):

Thank you for responding to my comments. However, there are still some questions that need to be addressed in the results section:

1. Results: What was the rationale for the selection of P0 and P3 for experimental time points? Do specific developmental processes occur in this window that could potentially make the time points differ?
2. Results: Similarly, how were the 3-week and 7-month time points selected? 3 weeks is the conventional weaning age; is it expected that weaning would affect the phenotype of interest?
3. Results: What was the goal of repeating the observational experiments in ICR mice? Is primordial follicle formation highly strain dependent?
4. Results: I would recommend the authors provide a brief written description of the overall experimental design to accompany Fig. 2e, rather than only the visual representation.
5. Results: The mitochondria-related mRNA levels in Fig. 3e show some differences between the P0 and P3 of the HFD group, while in most cases the P0 and P3 are largely comparable.

Reviewer #3 (Remarks to the Author):

In general, the authors have fully addressed my concerns.

Reviewer #4 (Remarks to the Author):

In this revised manuscript, the authors have addressed many of my concerns.

Q1. Thank you for providing the information about the women with GDM. It is notable that they did gain more weight during pregnancy than the controls, making this cohort more parallel with the mice with diet-induced weight gain during pregnancy.

Q2. Thank you for providing this new information.

Q3. I still think that the authors need to provide data for pregnancy weight gain, neonatal weight and litter size, as these are known phenotypes related to pregnancy exposures which can influence the health of F1 offspring. This information could be provided in the supplementary files so as not to distract from the abundance of other data, but is a required element for any papers reporting on phenotypes of offspring resulting from maternal exposures.

Q4. Ok, but some correction for metabolomics and other data should be performed. Alternatively, the term “nominal p value” should be used to allow the reader to understand that data are not adjusted.

Q5. Thank you for providing colors for Supplementary Figure 3D. It will be important in future studies to provide a spectrum of supplementation to confirm that levels are indeed related to endpoints.

**REVIEWER COMMENTS**

**Reviewer #1** (Remarks to the Author):

**The authors have sufficiently addressed all my concerns.**

**Response:** We appreciate the reviewer's approval of our revision.

**Reviewer #2** (Remarks to the Author):

**Thank you for responding to my comments. However, there are still some**

**questions that need to be addressed in the results section:**

**Q1: Results: What was the rationale for the selection of P0 and P3 for**
**experimental time points? Do specific developmental processes occur in this**
**window that could potentially make the time points differ?**

**Response:** We appreciate your professional comments. The selection of P0 and P3 as
experimental time points is grounded in the specific developmental processes, as
illustrated in Figure 1 a. Specifically, nest breakdown and primordial follicle assembly
commence around embryonic day 17.5 (E1 7.5), while the establishment of the
primordial follicle pool is completed shortly after birth, around P5 (Niu & Spradling,

2022). Analyzing the proportion of oocytes within follicles or nests during the E17.5-
P5 period is a widely recognized method of tracking the process of primordial follicle
formation (Lei & Spradling, 2013, 2016; Wang et al., 2020). By focusing on these time
points, our study aims to capture pivotal valuable insights into potential perturbations
that may occur during primordial follicle formation.

**Reference:**

- Niu, W., & Spradling, A. C. (2022). Mouse oocytes develop in cysts with the help of nurse cells. *Cell*,
*185(14)*, 2576-2590 e2512. doi:10.1016/j.cell.2022.05.001
- Lei, L., & Spradling, A. C. (2013). Female mice lack adult germ-line stem cells but sustain oogenesis
using stable primordial follicles. *Proc Natl Acad Sci U S A*, *110(21)*, 8585-8590.
doi:10.1073/pnas.1306189110
- Lei, L., & Spradling, A. C. (2016). Mouse oocytes differentiate through organelle enrichment from sister
cyst germ cells. *Science*, *352(6281)*, 95-99. doi:10.1126/science.aad2156
- Wang, J. J., Ge, W., Zhai, Q. Y., Liu, J. C., Sun, X. W., Liu, W. X., . . . Shen, W. (2020). Single-cell
transcriptome landscape of ovarian cells during primordial follicle assembly in mice. *PLoS Biol*,
*18(12)*, e3001025. doi:10.1371/journal.pbio.3001025

**Q2. Results: Similarly, how were the 3-week and 7-month time points selected? 3**
**weeks is the conventional weaning age; is it expected that weaning would affect**
**the phenotype of interest?**

**Response:** We appreciate the reviewer's attention to this aspect. The reduction in the
number of oocytes in follicle at P3 suggests that ovarian reserve might be impaired in
these mHFD offspring mice. In order to test this possibility, we analyzed the first wave
of follicular development in 3-week-old mice by counting the number of follicles at
various stages of development on the ovarian serial sections. The ovaries of 3-week-
old offspring from mHFD mice showed a decreased number of primordial follicles
(Fig.1h). Additionally, we further analyzed the fertility of mHFD offspring females by
crossing them to adult WT males. In a 5-month fertility test, mHFD offspring females
were subfertile and these 7-month-old mice show signs of premature ovarian failure.
The 3-week and 7-month time points were selected to assess the long-range effects of
maternal mHFD on offspring ovaries.

**Q3. Results: What was the goal of repeating the observational experiments in ICR**
**mice? Is primordial follicle formation highly strain dependent?**

**Response:** We appreciate the reviewer's attention to detail. Repeating the observational
experiments in ICR mice aimed to assess the generalizability of our findings. The
consistent phenotypic outcomes observed across diverse mouse strains indicate that
primordial follicle formation is not highly dependent on strain variations under the
conditions of maternal high-fat diet.

**Q4. Results: I would recommend the authors provide a brief written description**
**of the overall experimental design to accompany Fig. 2e, rather than only the**
**visual representation.**

**Response:** Thank you for your professional suggestions. To enhance the clarity of our
experimental design, we have provided a comprehensive description of the
experimental setup for Figure 2e in the subsection titled "VB1 deficiency and
supplementation models" at line 648 of the Methods section. Furthermore, in
accordance with your advice, we have supplemented the main text at line 141 with the
phrase "(Refer to the Methods for a detailed description of the model)" to guide readers
to refer to the Methods section for further elucidation.

**Q5. Results: The mitochondria-related mRNA levels in Fig. 3e show some different**
**between the PO and P3 of the HFD group, while in most cases the PO and P3 are**
**largely comparable.**

**Response:** We appreciate your professional review of our manuscript. The main focus
in Figure 3e is to compare the differences between the mHFD and mND groups at the
PO and P3 time points. To achieve this, we used PO_mND and P3 mND as references
for normalization, respectively, and compared the mHFD group with the mND group.
By using PO_mND as a reference for normalization across the four groups, the analysis
results indicate significant changes in mitochondrial-related mRNA levels in the
P3 mND group compared to the PO_mND group (as depicted in the figure below).
These findings confirm the dynamic changes in mitochondrial genes during the PO-P3

stage in the control group and suggest the potential importance of mitochondrial
 organization in the process of primordial follicle formation. Additionally, the result also
 demonstrates significant differences in mitochondrial-related genes between PO mND
 and PO mHFD, as well as between P3 mND and P3 mHFD. Notably, noticeable
 alterations in these genes can be observed both at PO and P3 in the HFD group.

 *Figure Legends: RT-qPCR analyzing the expression of mitochondrial organization related-*
 *genes in offspring ovaries from mND and mHFD groups (n > 6 biologically independent*
 *repeats). Statistical analyses were performed by one-way analysis of variance (ANOVA) with*
 *Tukey's test for multiple comparisons.*

 **Reviewer #3** (Remarks to the Author):

**In general, the authors have fully addressed my concerns.**

**Response:** Thank you for your approval of our revision.

 **Reviewer #4** (Remarks to the Author):

**In this revised manuscript, the authors have addressed many of my concerns.**

 **Q1. Thank you for providing the information about the women with GDM. It is**
 **notable that they did gain more weight during pregnancy than the controls,**
 **making this cohort more parallel with the mice with diet-induced weight gain**
 **during pregnancy.**

**Response:** Thank you for highlighting its potential implications.

**Q2. Thank you for providing this new information.**

**Response:** We appreciate your acknowledgment of the provided information.

**Q3. I still think that the authors need to provide data for pregnancy weight gain,**
**neonatal weight and litter size, as these are known phenotypes related to**
**pregnancy exposures which can influence the health of F1 offspring. This**
**information could be provided in the supplementary files so as not to distract from**
**the abundance of other data, but is a required element for any papers reporting**
**on phenotypes of offspring resulting from maternal exposures.**

**Response:** Thanks for your professional review on our manuscript. We have
incorporated these findings and the corresponding statistical analyses into the revised
manuscript (Supplementary Fig.4b-e), along with a more detailed description added in
line 142: *"At gestation day 19.5, both the mTA and mTD mice displayed no significant*
*changes in body weight compared to the mND mice (Supplementary Fig.4b, c). VB1*
*supplementation did not exert a significant effect on the body weight of pregnant*
*individuals in the mHFD mice (Supplementary Fig.4b, c). In addition, no statistically*
*significant disparities in litter size were observed across the experimental cohorts*
*(Supplementary Fig.4d)...."*

**Q4. Ok, but some correction for metabolomics and other data should be**
**performed. Alternatively, the term "nominal p value" should be used to allow the**
**reader to understand that data are not adjusted.**

**Response:** Thanks for your professional comments. Following your guidance, we have
incorporated the term "nominal p value" into our manuscript to enhance readers'
comprehension.

**QS. Thank you for providing colors for Supplementary Figure 3D. It will be**
**important in future studies to provide a spectrum of supplementation to confirm**
**that levels are indeed related to endpoints.**

**Response: We appreciate your acknowledgment of the color representation in**
**Supplementary Figure 3D. Your suggestions are very valuable for our future research.**

REVIEWERS' COMMENTS

Reviewer #2 (Remarks to the Author):

Thank you for further addressing the comments.

Reviewer #4 (Remarks to the Author):

The authors have adequately responded to my previous queries. Thank you.

**Manuscript ID:** NCOMMS-22-46558C

**Title:** Maternal vitamin B1 is a determinant for the fate of primordial follicle formation
in offspring.

**REVIEWER COMMENTS**

**Reviewer #2** (Remarks to the Author):

**Thank you for further addressing the comments.**

**Response:** Thank you for your approval of our revision.

**Reviewer #4** (Remarks to the Author):

**The authors have adequately responded to my previous queries. Thank you.**

**Response:** We appreciate the reviewer's approval of our revision.